# FireFlow: Fast Inversion of Rectified Flow for Image Semantic Editing

**Yingying Deng** [1]  **Xiangyu He** [2]  **Changwang Mei** [3]  **Peisong Wang** [2]  **Fan Tang** [4]

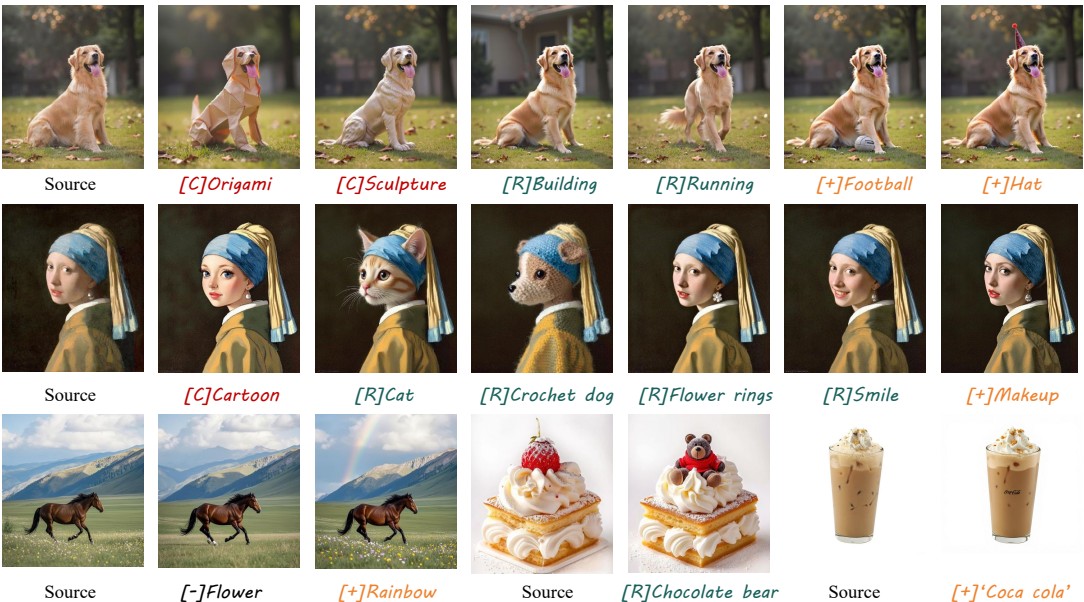

*Figure 1.* **FireFlow for Image Inversion and Editing in 8 Steps.** Our approach achieves outstanding results in semantic image editing and stylization guided by prompts, while maintaining the integrity of the reference content image and avoiding undesired alterations. [+]/[-] means adding or removing contents, [C] indicates changes in visual attributes (style, material, or texture), and [R] denotes content or gesture replacements.

## Abstract

Though Rectified Flows (ReFlows) with distillation offer a promising way for fast sampling, its fast inversion transforms images back to structured noise for recovery and following editing remains unsolved. This paper introduces FireFlow, an embarrassingly simple yet effective zero-shot approach that inherits the startling capacity of ReFlow-based models (such as FLUX) in generation while extending its capabilities to accurate inversion and editing in **8** steps. We first demonstrate that a carefully designed numerical solver is pivotal for ReFlow inversion, enabling accurate inversion and reconstruction with the precision of a second-order solver while maintaining the practical efficiency of a first-order Euler method. This solver achieves a $3\times$ runtime speedup compared to state-of-the-art ReFlow inversion and editing techniques while delivering smaller reconstruction errors and superior editing results in a training-free mode. The code is available at this-URL.

[1]University of Science and Technology Beijing, Beijing, China [2]Institute of Automation, Chinese Academy of Sciences, Beijing, China [3]Nanjing University of Science and Technology, Nanjing, China [4]Institute of Computing Technology, Chinese Academy of Sciences, Beijing, China. Correspondence to: Xiangyu He <hexiangyu17@mails.ucas.edu.cn>.

*Proceedings of the $42^{nd}$ International Conference on Machine Learning*, Vancouver, Canada. PMLR 267, 2025. Copyright 2025 by the author(s).

## 1. Introduction

The ability to accurately and efficiently invert generative models is critical for enabling applications such as semantic

image editing, data reconstruction, style transfer and latent space manipulation (Deng et al., 2024a; Hertz et al., 2022; Tumanyan et al., 2023; Deng et al., 2024b). Inversion, which involves mapping observed data back to its latent representation, serves as the foundation for fine-grained control over generative processes. Achieving a balance between computational efficiency and numerical accuracy in inversion is particularly challenging for diffusion models, which rely on iterative processes to bridge data and latent spaces.

Diffusion models have long been used for high-quality image generation (Ramesh et al., 2022; Rombach et al., 2022; Podell et al., 2024; Zhang et al., 2023; Deng et al., 2025) and inversion (Hertz et al., 2022; Parmar et al., 2023) due to their ability to capture complex distributions through stochastic differential equations (SDEs). Nevertheless, the nonlinear solver introduces undesirable drift in the reverse trajectory, thereby compromising the accuracy of reconstruction and the fidelity of editing-by-inversion. In contrast, rectified Flow (ReFlow) models (Liu et al., 2023), which replace stochastic sampling with ordinary differential equations (ODEs) for faster and more efficient transformations. This motivate a closer investigation of ReFlow-based models, particularly in the context of inversion and editing, to develop simple and effective methods.

ReFlow models possess an underutilized advantage: a well-trained ReFlow model learns nearly constant velocity dynamics across the data distribution, ensuring stability and bounded velocity approximation errors. However, existing inversion methods for ReFlow models fail to fully exploit this property (Rout et al., 2025; Wang et al., 2024). Current approaches rely on generic Euler solvers that prioritize each step's computational efficiency at the expense of accuracy or incur additional costs to achieve higher precision. As a result, the potential of ReFlow models to deliver fast and accurate inversion remains untapped.

In this work, we introduce a novel numerical solver for the ODEs underlying ReFlow models, addressing the challenges of inversion and editing. Our method achieves second-order precision while retaining the computational cost of a first-order solver. By reusing intermediate velocity approximations, our approach reduces redundant evaluations, stabilizes the inversion process, and fully leverages the constant velocity property of well-trained ReFlow models. As shown in Table 1, our approach is the first to provide a solver that strikes an optimal trade-off between accuracy and efficiency, enabling ReFlow models to excel in inversion and editing tasks. By combining computational efficiency, numerical robustness, and simplicity, our method offers a scalable solution for real-world tasks requiring high fidelity and real-time performance, advancing the utility of ReFlow-based generative models like FLUX.

*Table 1.* Comparison of recent training-free inversion and editing methods based on FLUX, including inversion/denoising steps, NFEs (Number of Function Evaluations) for both inversion and editing, local truncation error orders for solving ODE, and the need for a pre-trained auxiliary model for editing. Our approach offers a simple yet effective solution to address the challenges.

| Methods | Add-it | RF-Solver | RF-Inv. | Ours |
|---|---|---|---|---|
| **Steps** | 30 | 15 | 28 | 8 |
| **NFE** | 60 | 60 | 56 | 18 |
| **Aux. Model** | ✓ | w/o | w/o | w/o |
| **Local Error** | $\mathcal{O}(\Delta t^2)$ | $\mathcal{O}(\Delta t^3)$ | $\mathcal{O}(\Delta t^2)$ | $\mathcal{O}(\Delta t^3)$ |

(Tewel et al., 2025) for Add-it, (Wang et al., 2024) for RF-Solver, (Rout et al., 2025) for RF-Inv.

## 2. Preliminaries and Related Works

### 2.1. Rectified Flow

Rectified Flow (Liu et al., 2023) offers a principled approach for modeling transformations between two distributions, $\pi_0$ and $\pi_1$, based on empirical observations $X_0 \sim \pi_0$ and $X_1 \sim \pi_1$. The transformation is represented as an ordinary differential equation (ODE) over a continuous time interval $t \in [0, 1]$:

$$dZ_t = v(Z_t, t) \, dt, \tag{1}$$

where $Z_0 \sim \pi_0$ is initialized from the source distribution, and $Z_1 \sim \pi_1$ is generated at the end of the trajectory. The drift $v : \mathbb{R}^d \times [0, 1] \to \mathbb{R}^d$ is designed to align the trajectory of the flow with the direction of the linear interpolation path between $X_0$ and $X_1$. This alignment is achieved by solving the following least squares regression problem:

$$\min_v \ \mathbb{E} \left[ \int_0^1 \|(X_1 - X_0) - v_\theta(X_t, t)\|_2^2 \, dt \right], \tag{2}$$

where $X_t = tX_1 + (1-t)X_0$ denotes the linear interpolation path between $X_0$ and $X_1$.

**Forward process** seeks to transform samples $X_0 \sim \pi_0$ to match the target distribution $\pi_1$. A direct parameterization of $X_t$ is given by the linear interpolation $X_t = tX_1 + (1 - t)X_0$, which satisfies the non-causal ODE:

$$dX_t = (X_1 - X_0) \, dt. \tag{3}$$

However, this formulation assumes prior knowledge of $X_1$, rendering it non-causal and unsuitable for practical simulation. By introducing the drift $v(X_t, t)$, rectified flow causalizes the interpolation process. The drift $v$ is fit to approximate the linear direction $X_1 - X_0$, resulting in the forward ODE with $X_0 \sim \pi_0$,:

$$dX_t = v(X_t, t) \, dt, \quad t \in [0, 1]. \tag{4}$$

This causalized forward process enables simulation of $Z_t$ without requiring access to $X_1$ during intermediate time steps.

**Reverse process** generates samples from $\pi_1$ by reversing the learned flow. Starting from $X_1 \sim \pi_1$, the reverse ODE is given by negating the drift term:

$$dX_t = -v(X_t, t)\, dt, \quad t \in [1, 0]. \qquad (5)$$

This process effectively "undoes" the transformations applied during the forward flow, enabling the generation of $X_0$ that follows the original distribution $\pi_0$. The reverse process guarantees consistency with the forward dynamics by leveraging the symmetry of the learned drift $v$.

## 2.2. Inversion

The inversion of real images into noise feature space, as well as the reconstruction of noise features back to the original real images, is a prominent area of research within diffusion models applied to image editing tasks (Lin et al., 2024; Brack et al., 2024; Miyake et al., 2023; Ju et al., 2024; Zhang et al., 2022; Huberman-Spiegelglas et al., 2024; Cho et al., 2024). The foundational theory of Denoising Diffusion Implicit Models (DDIM) (Song et al., 2021) involves the addition of predicted noise to a fixed noise during the forward process, which can subsequently be mapped to generate an image. However, this approach encounters challenges related to reconstruction bias. To address this issue, Null-Text Inversion (Mokady et al., 2023) optimizes an input null-text embedding, thereby correcting reconstruction errors at each iterative step. Similarly, Prompt-Tuning-Inversion (Dong et al., 2023) refines conditional embeddings to accurately reconstruct the original image. Negative-Prompt-Inversion (Miyake et al., 2023) replaces the null-text embedding with prompt embeddings to expedite the inversion process. Direct-Inversion (Ju et al., 2024) incorporates the inverted noise corresponding to each timestep within the denoising process to mitigate content leakage.

In contrast to the aforementioned Stochastic Differential Equation (SDE)-based formulations, rectified flow models that utilize ordinary differential equations (ODEs) offer a more direct solution pathway. RF-Inversion (Rout et al., 2025) employs dynamic optimal control techniques derived from linear quadratic regulators, while RF-Solver (Wang et al., 2024) utilizes Taylor expansion to minimize inversion errors in ODEs. Nevertheless, achieving superior inversion results typically necessitates an increased number of generation steps, which can lead to significant computational time and resource expenditure. In this paper, we propose a few-step ODE solver designed to balance effective outcomes with high efficiency.

## 2.3. Editing

Image editing utilizing a pre-trained diffusion model has demonstrated promising results, benefiting from advance-

ments in image inversion and attention manipulation technologies (Hertz et al., 2022; Cao et al., 2023; Meng et al., 2022; Couairon et al., 2023; Deutch et al., 2024; Xu et al., 2024; Brooks et al., 2023). Training-free editing methods typically employ a dual-network architecture: one network is dedicated to reconstructing the original image, while the other is focused on editing. Prompt-to-Prompt (Hertz et al., 2022) manipulates the cross-attention maps within the editing pipeline by leveraging features from the reconstruction pipeline. Plug-and-Play (Tumanyan et al., 2023) method substitutes the attention matrices of self-attention blocks in the editing pipeline with those from the reconstruction pipeline. Similarly, MasaCtrl (Cao et al., 2023) modifies the Value components of self-attention blocks in the editing pipeline using values derived from the reconstruction pipeline. Additionally, the Add-it (Tewel et al., 2025) method utilizes both the Key and Value components of self-attention blocks from the source image to guide the editing process effectively. With respect to FLUX models, RF-Editing (Wang et al., 2024) reuses Value components from MMDiT (generated during inversion) for reconstruction-based editing, while RF-Inversion (Rout et al., 2025) uses the original image as a prior for guiding the forward and reverse process in semantic editing. We introduce FireFlow, a novel solver that can effectively integrate these techniques (and can also be used with the vanilla ReFlow model) to achieve promising results with fewer steps.

## 3. Motivation

The ReFlow model operates under the simple assumption that $X_t$ evolves linearly between $X_0$ and $X_1$, corresponding to uniform linear motion. Drawing an analogy to physics, it is natural to extend this linear motion to accelerated motion by incorporating an acceleration term:

$$\frac{dv_t}{dt} = a(X_t, t), \quad \frac{dX_t}{dt} = v(X_t, t), \qquad (6)$$

where $X_{t+1} = X_t + v_t \Delta t + \frac{1}{2} a_t \Delta t^2$, and $v_t$ is equivalent to $v(X_t, t)$ for simplicity. Recent works have empirically shown that training-based strategies (Park et al., 2024; Chen et al., 2024) for solving Equation (6) improve coupling preservation and inversion over rectified flow, even with few steps. Moreover, a training-free method (Wang et al., 2024) leveraging pre-trained ReFlow models has also demonstrated the utility of the second-order derivative of $v$ in achieving effective inversion, essentially aligning with the principles of accelerated motion.

However, this observation appears counterintuitive. A well-trained ReFlow model, such as FLUX, generally assumes that $v_t$ approximates the constant value $X_1 - X_0$. Thus, the acceleration term $a_t = dv_t/dt$ theoretically approaches zero, as the learning target for $v_t$ is constant.

**Connection to High-Order ODE Solvers:** Instead of treating $a_t$ as a continuous term, we reinterpret it through the lens of high-order ODE solvers. Using the finite-difference approximation $a_t = (v_{t+\Delta t} - v_t)/\Delta t$, we can rewrite the equation as:

$$X_{t+1} = X_t + v_t\Delta t + \frac{1}{2}a_t\Delta t^2 = X_t + \frac{1}{2}(v_t + v_{t+\Delta t})\Delta t,$$

which corresponds to the standard formulation of the second-order Runge-Kutta method. This high-order approach allows fewer steps (or equivalently, larger step sizes $\Delta t$) to achieve the same accuracy as Euler's method, since the global error of a $p$-th order method scales as $\mathcal{O}(\Delta t^p)$. This enables larger $\Delta t$ while maintaining the same error tolerance $\epsilon$. Similarly, if we approximate $a_t$ using $a_t = (v_{t+\frac{1}{2}\Delta t} - v_t)/(\frac{1}{2}\Delta t)$, the resulting position update becomes:

$$X_{t+1} = X_t + v_{t+\frac{1}{2}\Delta t}\Delta t,$$

which corresponds to the standard midpoint method, another second-order ODE solver.

**Impact on ReFlow Inversion:** It is well-established that the global error in the forward process of ODE solvers benefits from higher-order methods. Likewise, inversion and reconstruction tasks also exhibit improved performance with high-order solvers, as they better preserve original image details during the inversion of ReFlow models. We formalize this property in the following statement.

**Proposition 3.1.** *Given a $p$-th order ODE solver and the ODE $\frac{dX_t}{dt} = v_\theta(X_t, t)$, if the dynamics of the reverse pass satisfy $\frac{dX_t}{dt} = -v_\theta(X_t, 1-t)$ which is Lipschitz continuous with constant L. The perturbation $\Delta_T$ at $t = T$ propagates backward to $t = 0$. The propagated error satisfies:*

$$\|\Delta_0\| \leq e^{-LT}\|\Delta_T\|. \tag{7}$$

**Implication.** The inversion error $\|\Delta_T\|$ introduced during the $p$-th order numerical solution propagates into the reverse pass, experiencing a slight reduction scaled by the Lipschitz constant $L$ of the learned drift $v_\theta(X_t, t)$. Despite this reduction, the overall reconstruction error $\|\Delta_0\|$ for the original image remains asymptotically of the same order, $\mathcal{O}(\Delta t^p)$, where $\Delta t$ represents the integration step size. Consequently, high-order solvers are preferred in ReFlow to achieve accurate inversion and editing with fewer steps.

## 4. Method

**Challenges with High-Order Solvers:** While the use of high-order solvers is theoretically promising, it fails to yield practical runtime speedups. For a parameterized drift $v_\theta(X_t, t)$, the runtime is determined by the Number of Function Evaluations (NFEs), i.e., the number of forward passes through the model $v_\theta(X_t, t)$. High-order solvers require

evaluating more points within the interval $[t, t+1]$, leading to a higher NFE per step, which negates any reduction in the number of steps and fails to improve overall computational efficiency.

For instance, the midpoint method achieves a local error of $\mathcal{O}(\Delta t^3)$ and a global error of $\mathcal{O}(\Delta t^2)$. Formally, it proceeds as follows:

$$X_{t+\frac{\Delta t}{2}} = X_t + \frac{\Delta t}{2}v_\theta(X_t, t), \tag{8}$$

$$X_{t+1} = X_t + \Delta t \cdot v_\theta(X_{t+\frac{\Delta t}{2}}, t + \frac{\Delta t}{2}). \tag{9}$$

This scheme requires two NFEs per step: one to compute $X_{t+\frac{\Delta t}{2}}$ and another for $v_\theta(X_{t+\frac{\Delta t}{2}}, t + \frac{\Delta t}{2})$, effectively doubling the cost compared to the Euler method. The midpoint method leverages $v_{t+\frac{\Delta t}{2}}$ to provide a more accurate estimate of $\frac{X_{t+1}-X_t}{\Delta t}$ than $v_t$, which inspires us to seek an alternative with lower computational cost.

**A Low-Cost Alternative:** The training objective of Re-Flow implies that a well-trained model satisfies $v_\theta(X_t, t) \approx (X_1 - X_0)$ for all $t$. Leveraging this property, the most efficient approach would replace $v_t$ with $v_0$, enabling one-step generation as proposed in the original ReFlow method (Liu et al., 2023). However, this simplification makes it difficult to incorporate conditional priors, as multi-step iteration is no longer required.

To maintain a multi-step paradigm, we propose a modified scheme that replaces $v_t$ with previous $t - 1$-step midpoint velocity $v_{(t-1)+\frac{\Delta t}{2}}$ rather than $v_{t+\frac{\Delta t}{2}}$. This approach is formalized as:

$$\hat{v}_\theta(X_t, t) \coloneqq \underbrace{v_\theta(X_{(t-1)+\frac{\Delta t}{2}}, (t-1) + \frac{\Delta t}{2})}_{\text{load from memory}} \tag{10}$$

$$\hat{X}_{t+\frac{\Delta t}{2}} \coloneqq X_t + \frac{\Delta t}{2}\hat{v}_\theta(X_t, t) \tag{11}$$

$$X_{t+1} = X_t + \Delta t \cdot \underbrace{v_\theta(\hat{X}_{t+\frac{\Delta t}{2}}, t + \frac{\Delta t}{2})}_{\text{run \& save to memory}} \tag{12}$$

In this simple scheme, only one NFE is required per step[1], matching the computational cost of the Euler method. The key question, then, is whether this scheme retains the second-order accuracy of the original midpoint method.

For the local and global truncation error, we derive that if $v_\theta(X_t, t)$ is well-trained and varies smoothly with respect to both $X$ and $t$, the proposed scheme achieves the same truncation error as the standard midpoint method. This ensures that the modified approach retains the benefits of

---

[1]Specifically, we perform two NFEs at $t = 0$ to initialize the conditions $v_0$ and $v_{0+\frac{\Delta t}{2}}$ for subsequent iterations. Python-style pseudo-code is provided in Sec.D.

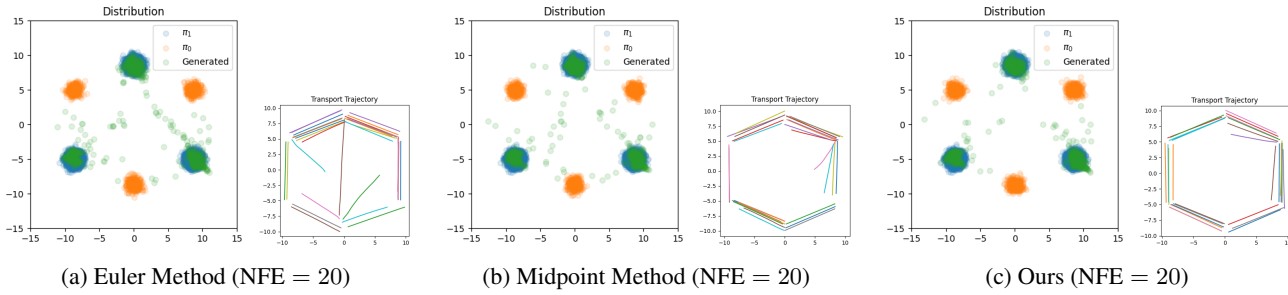

(a) Euler Method (NFE = 20)  (b) Midpoint Method (NFE = 20)  (c) Ours (NFE = 20)

*Figure 2.* Results on 2D synthetic dataset. We evaluate the performance of 2-Rectified Flow using the Euler solver, midpoint solver, and our proposed approach on a 2D synthetic dataset. The source distribution $\pi_0$ (orange) and the target distribution $\pi_1$ (green) are parameterized as Gaussian mixture models. For the Euler method, the number of sampling steps is set to $N = 20$, corresponding to an NFE of 20. Our approach generates samples that align more closely with the target distribution, achieving a better match in density and structure. Additionally, the trajectories of the samples exhibit greater straightness, adhering closely to the ideal of linear motion.

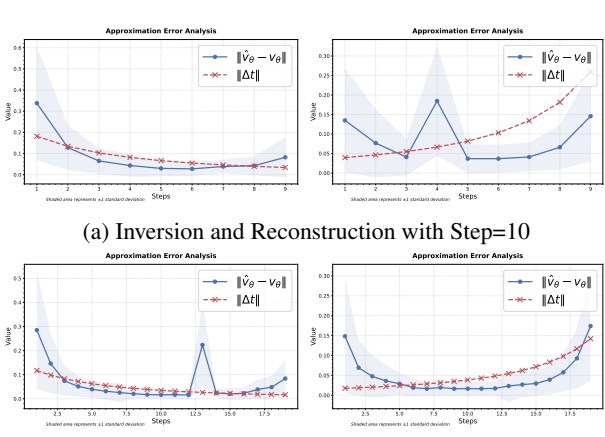

(a) Inversion and Reconstruction with Step=10

(b) Inversion and Reconstruction with Step=20

*Figure 3.* Illustrations of the approximation error in velocity ($\|\hat{v}_\theta - v_\theta\|$) as it evolves with inversion steps (left subfigures) and denoising steps (right subfigures), with $\Delta t$ included as a reference.

second-order accuracy while operating at the computational cost of a first-order solver.

**Proposition 4.1.** *Let $\hat{v}_\theta(X_t, t)$ denote the reused velocity approximation in Equation 10, and $v_\theta(X_t, t)$ denotes the exact velocity at time $t$. Then, the approximation satisfies the error bound: $\|\hat{v}_\theta(X_t, t) - v_\theta(X_t, t)\| \leq \mathcal{O}(\Delta t)$, under the following conditions:*

*1) Temporal Error: The temporal error is directly proportional to the time step $\Delta t$, stemming from smoothness of $v_\theta(X, t)$ in the time domain.*

*2) Spatial Error: The spatial error is dominated by $\mathcal{O}(\Delta t)$, due to the boundedness of $\frac{\partial v_\theta}{\partial X}$.*

We formally prove in the appendix that when two required conditions are satisfied, leading to the following result: our modified midpoint method achieves the same truncation error as the standard midpoint method under these circum-

stances. Consequently, it is expected to exhibit smaller overall error while maintaining the same runtime cost as the first-order Euler method.

**Theorem 4.2.** *Consider a ReFlow model governed by ODE: $\frac{dX}{dt} = v_\theta(X, t)$, where $v_\theta(X, t)$ is smooth and bounded, and the solution $X_t$ evolves over a time interval $[0, T]$. The modified midpoint method, defined in Equation (12), achieves the same global truncation error $\mathcal{O}(\Delta t^2)$ as the standard midpoint method, provided the reused velocity satisfies: $\|\hat{v}_\theta(X_t, t) - v_\theta(X_t, t)\| \leq \mathcal{O}(\Delta t)$.*

To highlight the advantages of our approach, we conduct experiments on synthetic data following the setup in (Liu et al., 2023). As shown in Figure 2, the transport trajectories generated by our method are straighter, leading to improved accuracy while maintaining the same NFE as the Euler method, and even surpassing the performance of the standard midpoint method.

**Numerical Results and Discussion** To empirically validate the theoretical assumption that the reused velocity approximation error $\|\hat{v}_\theta(X_t, t) - v_\theta(X_t, t)\|$ is bounded by $\mathcal{O}(\Delta t)$, we conducted numerical experiments and analyzed the relationship between the approximation error and the time step size $\Delta t$ on FLUX-dev model during inversion and reconstruction. The results are summarized in Figure 3, which depicts the average approximation error across different step sizes, with the shaded area representing $\pm 1$ standard deviation.

The data exhibit the following key trend that the approximation error grows approximately linearly with the step size $C \cdot \Delta t$, consistent with the theoretical bound $\mathcal{O}(\Delta t)$. Despite the occasional spikes and inherent variability of the error (illustrated by the shaded standard deviation), the trend and magnitude of the error remain well-controlled and stable across most steps, further validating the robustness of the reused velocity approximation in practice.

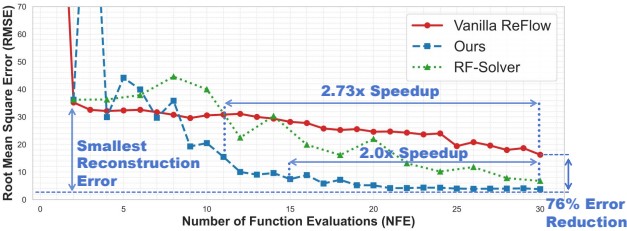

*Figure 4.* Image reconstruction errors versus denoising NFE: Our approach, compared to the first-order vanilla ReFlow inversion and second-order RF-solver, achieves lower reconstruction errors and demonstrates faster convergence with respect to NFE.

**Image Semantic Editing:** To ensure simplicity and a fair comparison with the second-order solver method, we follow the approach in (Wang et al., 2024), where the value features in self-attention layers during the denoising process are replaced with pre-stored value features generated during the inversion process, serving as a prior. Subsequently, a reference prompt is used as guidance to achieve semantic editing. Leveraging superior image preservation, our method does not require a careful selection of timesteps for applying the replacements, as suggested in the original paper. The inversion and denoising sampling processes are detailed in Algorithms 1 and 2.

## 5. Experiment

### 5.1. Implementation Details

**Baselines:** This section compares FireFlow with DM inversion-based editing methods such as Prompt-to-Prompt (Hertz et al., 2022) , MasaCtrl (Cao et al., 2023), Pix2Pix-zero (Parmar et al., 2023), Plug-and-Play (Tumanyan et al., 2023), DiffEdit (Couairon et al., 2023) and DirectInversion (Ju et al., 2024). We also consider the recent RF inversion methods, such as RF-Inversion (Rout et al., 2025) and RF-Solver (Wang et al., 2024).

**Metrics:** We evaluate different methods across three aspects: generation quality, text-guided quality, and preservation quality. The Fréchet Inception Distance (FID) (Heusel et al., 2017) is used to measure image generation quality by comparing the generated images to real ones. A CLIP model (Radford et al., 2021) is used to calculate the similarity between the generated image and the guiding text. To assess the preservation quality of non-edited areas, we use metrics including Learned Perceptual Image Patch Similarity (LPIPS) (Zhang et al., 2018), Structural Similarity Index Measure (SSIM) (Wang et al., 2004), Peak Signal-to-Noise Ratio (PSNR), and structural distance (Ju et al., 2024).

**Steps:** Since the number of inference steps can significantly impact performance, we follow the best settings reported for the RF-Solver to ensure a fair comparison: 10 steps for text-

*Table 2.* Quantitative results for Unconditional Image Generation on CIFAR-10. Our approach achieves performance comparable to that of a second-order solver, but with reduced computational cost.

| ODE Solver | Steps(↓) | NFE(↓) | FID(↓) | IS(↑) |
|---|---|---|---|---|
| **Euler** | 15 | 15 | 5.67 | 9.20 |
| **Euler** | 10 | 10 | 5.83 | 9.03 |
| **Midpoint** | 5 | 10 | 5.45 | **9.27** |
| **Ours** | 5 | 6 | **5.35** | 9.26 |

*Table 3.* Quantive results on Text-to-Image Generation. The FLUX-dev uses 1st-order ODE Solver, the RF-Solver and our approach use 1st-order ODE Solver. Compared with FLUX-dev and RF-Solver, our approach has higher FID and CLIP Score.

| Methods | Steps | NFE(↓) | FID(↓) | CLIP Score(↑) |
|---|---|---|---|---|
| **FLUX-dev** | 20 | 20 | 26.77 | **31.44** |
| **RF-Solver** | 20 | 40 | 25.54 | 31.39 |
| **RF-Solver** | 10 | 20 | 25.93 | 31.35 |
| **Ours** | 10 | 11 | **25.16** | 31.42 |

to-image generation (T2I) and 30 steps for reconstruction. For editing, RF-Solver varies the number of steps by task, using up to 25 steps. In contrast, we find that our approach achieves comparable or better results using 8 steps. The ablation study is shown in Section E.

### 5.2. Text-to-image Generation

**Conditional Image Generation:** We compare the performance of our method against the vanilla rectified flow and the second-order RF-solver on the fundamental T2I task. Following the setup in RF-solver, we evaluate a randomly selected subset of 10K images from the MSCOCO Caption 2014 validation set (Chen et al., 2015), using the ground-truth captions as reference prompts. The FID and CLIP scores for results generated with a fixed random seed of 1024 are presented in Table 3. In summary, our method delivers superior image quality while maintaining comparable text alignment performance.

**Unconditional Image Generation:** We adhere to the protocol established in the original ReFlow paper (Liu et al., 2023) for unconditional image generation on CIFAR-10, utilizing the open-source 1-Rectified-Flow-distill weights. The results, as shown in Table 2, compare various ODE solvers, and our method demonstrates consistent performance with fewer steps and higher scores.

### 5.3. Inversion and Reconstruction

**Quantitative Comparison:** We report the inversion and reconstruction results on the first 1K images from the Densely Captioned Images (DCI) dataset (Urbanek et al., 2024), using the official descriptions as source prompts. The results,

*Table 4.* Compare our approach with other editing methods on PIE-Bench. Our method demonstrates superior performance in terms of background preservation and CLIP similarity.

| Method | Model | Structure | Background Preservation | | CLIP Similarity↑ | | Steps | NFE↓ |
|---|---|---|---|---|---|---|---|---|
| | | Distance↓ | PSNR↑ | SSIM↑ | Whole | Edited | | |
| Prompt2Prompt (Hertz et al., 2022) | Diffusion | 0.0694 | 17.87 | 0.7114 | 25.01 | 22.44 | 50 | 100 |
| Pix2Pix-Zero (Parmar et al., 2023) | Diffusion | 0.0617 | 20.44 | 0.7467 | 22.80 | 20.54 | 50 | 100 |
| MasaCtrl (Cao et al., 2023) | Diffusion | 0.0284 | 22.17 | 0.7967 | 23.96 | 21.16 | 50 | 100 |
| PnP (Tumanyan et al., 2023) | Diffusion | 0.0282 | 22.28 | 0.7905 | 25.41 | 22.55 | 50 | 100 |
| PnP-Inv. (Ju et al., 2024) | Diffusion | **0.0243** | 22.46 | 0.7968 | 25.41 | 22.62 | 50 | 100 |
| RF-Inversion (Rout et al., 2025) | ReFlow | 0.0406 | 20.82 | 0.7192 | 25.20 | 22.11 | 28 | 56 |
| RF-Solver (Wang et al., 2024) | ReFlow | 0.0311 | 22.90 | 0.8190 | 26.00 | 22.88 | 15 | 60 |
| Ours | ReFlow | 0.0284 | **23.29** | **0.8287** | 26.01 | **22.96** | 15 | 32 |
| Ours | ReFlow | 0.0271 | 23.03 | 0.8249 | **26.02** | 22.81 | **8** | **18** |

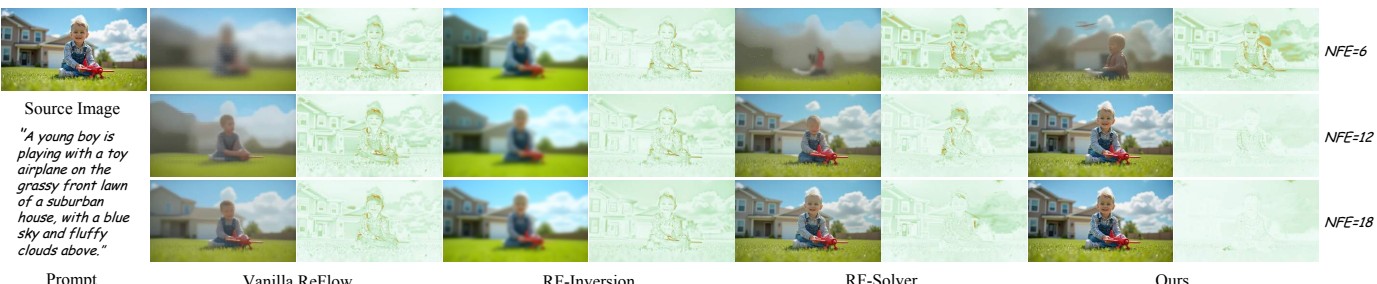

*Figure 5.* Qualitative results of image reconstruction. Our approach achieves faster convergence and superior reconstruction quality compared to baseline ReFlow methods utilizing the FLUX model. Difference images showing the pixel-wise variations between the source image and the reconstructed images are also provided.

*Table 5.* Quantitative results for inversion and reconstruction using the FLUX-dev model (excluding the DDIM baseline). NFE includes both inversion and reconstruction function evaluations. Steps or computational costs are kept comparable across comparisons. **Reconstruction is performed without leveraging latent features from the inversion process.**

| | Steps | NFE↓ | LPIPS↓ | SSIM↑ | PSNR↑ |
|---|---|---|---|---|---|
| DDIM-Inv. | 50 | 100 | 0.2342 | 0.5872 | 19.72 |
| ReFlow-Inv. | 30 | 60 | 0.5044 | 0.5632 | 16.57 |
| RF-Solver | 30 | 120 | 0.2926 | 0.7078 | 20.05 |
| RF-Inversion | 30 | 60 | 0.4480 | 0.4599 | 16.22 |
| Ours | 30 | 62 | **0.1579** | **0.8160** | **23.87** |
| ReFlow-Inv. | 9 | 18 | 0.8145 | 0.3828 | 15.29 |
| RF-Solver | 5 | 20 | 0.5010 | 0.5232 | 14.72 |
| RF-Inversion | 9 | 18 | 0.7574 | 0.3742 | 13.37 |
| Ours | 8 | 18 | **0.4111** | **0.5945** | **16.01** |

shown in Table 5, demonstrate that our approach achieves a significant reduction in reconstruction error, whether compared at the same number of steps (yielding approximately $2\times$ speedup) or at the same computational cost.

**Qualitative Comparison:** As shown in Figure 5, our approach provides an efficient and effective reconstruction method based on FLUX. The drift from the source image is

significantly smaller compared to baseline methods, aligning with the quantitative results.

**Convergence Rate:** We empirically compare the convergence rates of different numerical solvers during reconstruction, as shown in Figure 4. For a fair comparison, we use the demo "boy" image and prompt provided in the RF-solver source code. Our approach achieves the lowest reconstruction error with the fastest convergence rate, offering up to $2.7\times$ speedup and over $70\%$ error reduction. Figure 7 further illustrates the results when other approaches are fully converged at denoising NFE $= 60$.

### 5.4. Inversion-based Semantic Image Editing

**Quantitative Comparison:** We evaluate prompt-guided editing using the recent PIE-Bench dataset (Ju et al., 2024), which comprises 700 images across 10 types of edits. As shown in Table 4, we compare the editing performance in terms of preservation ability and CLIP similarity. Our method not only competes with but often outperforms other approaches, particularly in CLIP similarity. Notably, our approach achieves high-quality results with relatively few editing steps, demonstrating its efficiency and effectiveness in maintaining the integrity of the original content while producing edits that closely align with the intended modifi-

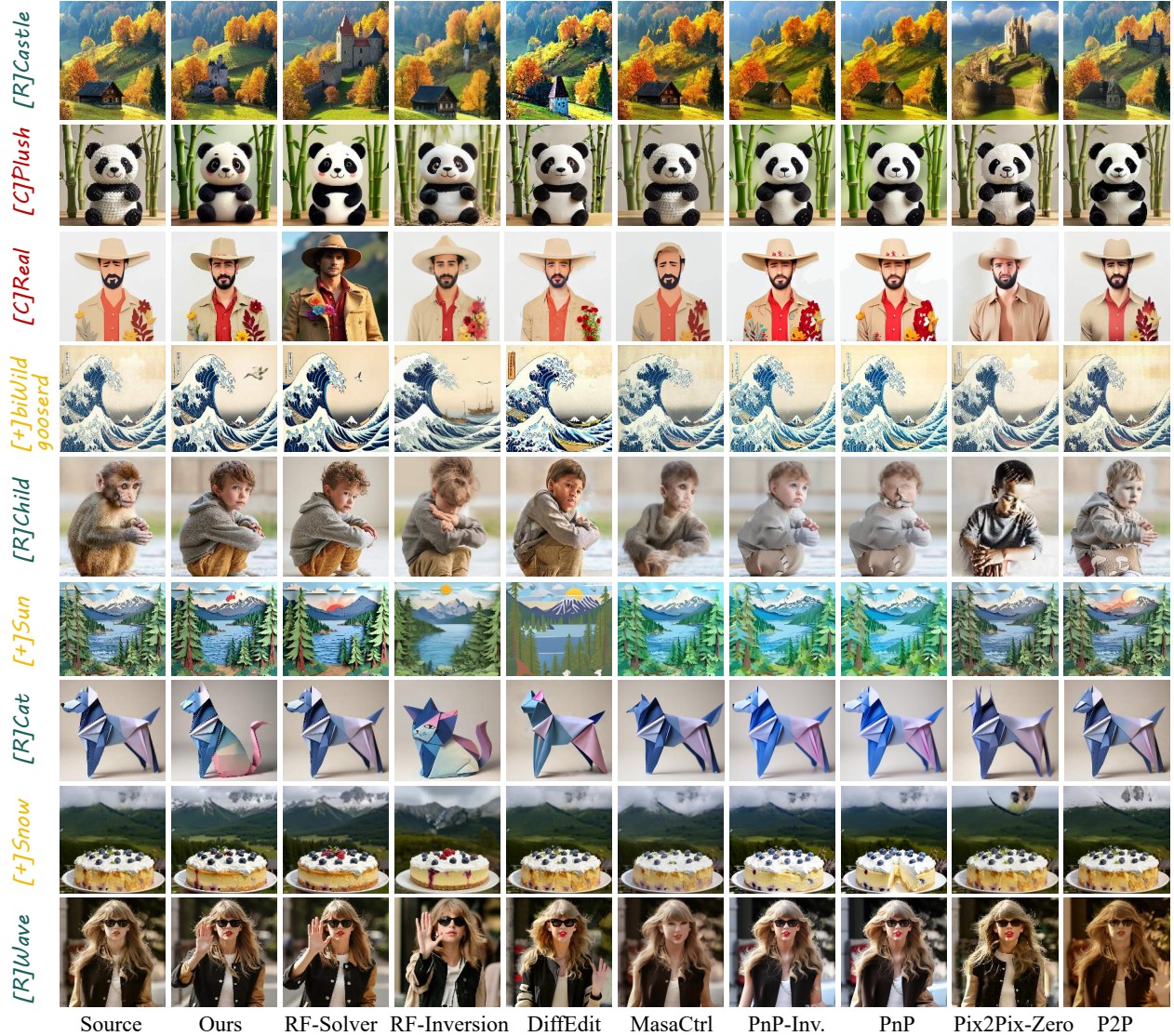

*Figure 6.* Comparison with State-of-the-art editing methods. As shown in the comparison (original images in Row 1 vs. edited results in subsequent rows), our approach delivers the most accurate and high-quality edits.

cations.

**Qualitative Comparison:** We present the visual editing results in Figure 6, which are consistent with our quantitative findings. Our method highlights a fundamental trade-off between minimizing changes to non-editing areas and enhancing the fidelity of the edits. In contrast, methods like P2P, Pix2Pix-Zero, MasaCtrl, and PnP often struggle with inconsistencies relative to the source image, as evident in the 3rd and 6th rows of the figure. Additionally, these methods frequently produce invalid edits, as shown in the 7th and 9th rows. While PnP-Inv excels at preserving the structure of the source image, it often fails to effectively apply the desired edits. Rectified flow model-based methods, such as RF-Inversion and RF-Solver, deliver better editing re-

*Table 6.* Per-image inference time for ReFlow inversion-based editing measured on an RTX 3090.

|  | Resolution | Steps | Time Cost | Speedup |
|---|---|---|---|---|
| Vanilla ReFlow | $512 \times 512$ | 28 | 23.76s | 1.0× |
| RF-Inversion | $512 \times 512$ | 28 | 23.36s | 1.02× |
| RF-Solver | $512 \times 512$ | 15 | 25.31s | 0.94× |
| Ours | $512 \times 512$ | 8 | **7.70**s | **3.09**× |
| Vanilla ReFlow | $1024 \times 1024$ | 28 | 72.10s | 1.0× |
| RF-Inversion | $1024 \times 1024$ | 28 | 71.35s | 1.01× |
| RF-Solver | $1024 \times 1024$ | 15 | 78.80s | 0.92× |
| Ours | $1024 \times 1024$ | 8 | **24.52**s | **2.94**× |

sults compared to the aforementioned methods. However, they still face challenges with inconsistencies in non-editing areas. Overall, our method provides a more effective solution to these challenges, achieving superior results in both preservation and editing fidelity.

**Inference Speed:** In Table 6, we compare the inference time of FireFlow with several recent models. The number of steps is based on those reported in the original papers or provided in open-source code. FireFlow is significantly faster than competing reflow models and does not require an auxiliary model for editing.

## 6. Conclusion

We proposed a novel numerical solver for ReFlow models, achieving second-order precision at the computational cost of a first-order method. By reusing intermediate velocity approximations, our training-free approach fully exploits the nearly constant velocity dynamics of well-trained ReFlow models, minimizing computational overhead while maintaining accuracy and stability. This method addresses key limitations of existing inversion techniques, providing a scalable and efficient solution for tasks such as image reconstruction and semantic editing. Our work highlights the untapped potential of ReFlow-based generative frameworks and establishes a foundation for further advancements in efficient numerical methods for generative ODEs. We also provide a discussion of the limitations in Section F.

## Acknowledgement

This work was partially supported by the National Key R&D Program of China (No. 2022ZD0160304), the Beijing Science and Technology Plan Project (No. Z231100005923033) and CCF-Tencent Rhino-Bird Open Research Fund.

## Impact Statement

This paper presents work whose goal is to advance the field of Machine Learning. There are many potential societal consequences of our work, none of which we feel must be specifically highlighted here.

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

# A. The Pseudo-code for Inversion and Editing

---

**Algorithm 1** Solving ReFlow Inversion ODE

---

**Require:** Discretization steps $N$, reference image $X_0$, prompt embedding network $\Phi$, Flux model $v(\cdot, \cdot, \cdot; \varphi)$, time steps $t = [t_0, ..., t_{N-1}]$

**Ensure:** Structured noise $X_1$

1: Initialize $v_{t_0}(X_{t_0}) = v(X_{t_0}, t_0, \Phi(\cdot); \varphi)$        {Run}

2: $\Delta t_0 = t_1 - t_0$

3: $X_{t_0 + \frac{1}{2}\Delta t_0} = X_{t_0} + \frac{1}{2}\Delta t_0 \cdot v_{t_0}(X_{t_0})$

4: Initialize $v_{t_0 + \frac{1}{2}\Delta t_0}(X_{t_0 + \frac{1}{2}\Delta t_0}) = v(X_{t_0 + \frac{1}{2}\Delta t}, t_0 + \frac{1}{2}\Delta t_0, \Phi(\cdot); \varphi)$        {Run & Save to GPU Memory}

5: $X_{t_1} = X_{t_0} + \Delta t_0 \cdot v_{t_0 + \frac{1}{2}\Delta t_0}(X_{t_0 + \frac{1}{2}\Delta t_0})$

6: **for** $i = 1 : N - 1$ **do**

7:     $\hat{v}_{t_i}(X_{t_i}) \leftarrow v_{t_{i-1} + \frac{1}{2}\Delta t_{i-1}}(X_{t_{i-1} + \frac{1}{2}\Delta t_{i-1}})$        {Load from GPU Memory}

8:     $\Delta t_i = t_{i+1} - t_i$

9:     $X_{t_i + \frac{1}{2}\Delta t_i} = X_{t_i} + \frac{1}{2}\Delta t_i \cdot \hat{v}_{t_i}(X_{t_i})$

10:     $v_{t_i + \frac{1}{2}\Delta t_i}(X_{t_i + \frac{1}{2}\Delta t_i}) = v(X_{t_i + \frac{1}{2}\Delta t_i}, t_i + \frac{1}{2}\Delta t_i, \Phi(\cdot); \varphi)$        {Run & Save to GPU Memory}

11:     $X_{t_{i+1}} = X_{t_i} + \Delta t_i \cdot v_{t_i + \frac{1}{2}\Delta t_i}(X_{t_i + \frac{1}{2}\Delta t_i})$

12:     **if** i==$N - 1$ **then**

12:         Save $V_{t_{N-1}}^{inv.}$ to storage.

13:     **end if**

14: **end for**

15: **return** $X_1, V_{t_{N-1}}^{inv.}$ in Self-attention Layers

---

**Algorithm 2** Solving ReFlow Denoising ODE (Editing)

---

**Require:** Discretization steps $N$, reference text "prompt", structured noise $X_1$, prompt embedding network $\Phi$, Flux model $v(\cdot, \cdot, \cdot; \varphi)$, time steps $t = [t_{N-1}, ..., t_0]$, pre-computed $V_{t_{N-1}}^{inv.}$ in Self-attention Layers during inversion

**Ensure:** Edited image $X_0$

1: Initialize $v_{t_{N-1}}(X_{t_{N-1}}) = v(X_{t_{N-1}}, t_{N-1}, \Phi(\text{prompt}); \varphi)$     {Replace $V_{t_{N-1}}^{edit}$ with $V_{t_{N-1}}^{inv.}$ in Self-attention & Run}

2: $\Delta t_{N-1} = t_{N-2} - t_{N-1}$

3: $X_{t_{N-1} + \frac{1}{2}\Delta t_{N-1}} = X_{t_{N-1}} + \frac{1}{2}\Delta t_{N-1} \cdot v_{t_{N-1}}(X_{t_{N-1}})$

4: Initialize $v_{t_{N-1} + \frac{1}{2}\Delta t_{N-1}}(X_{t_{N-1} + \frac{1}{2}\Delta t_{N-1}}) = v(X_{t_{N-1} + \frac{1}{2}\Delta t_{N-1}}, t_{N-1} + \frac{1}{2}\Delta t_{N-1}, \Phi(\text{prompt}); \varphi)$ {Replace $V_{t_{N-1}}^{edit}$ with $V_{t_{N-1}}^{inv.}$ in Self-attention & Run & Save to GPU Memory}

5: $X_{t_{N-2}} = X_{t_{N-1}} + \Delta t_{N-1} \cdot v_{t_{N-1} + \frac{1}{2}\Delta t_{N-1}}(X_{t_{N-1} + \frac{1}{2}\Delta t_{N-1}})$

6: **for** $i = N - 2 : 0$ **do**

7:     $\hat{v}_{t_i}(X_{t_i}) \leftarrow v_{t_{i+1} + \frac{1}{2}\Delta t_{i+1}}(X_{t_{i+1} + \frac{1}{2}\Delta t_{i+1}})$        {Load from GPU Memory}

8:     $\Delta t_i = t_{i-1} - t_i$

9:     $X_{t_i + \frac{1}{2}\Delta t_i} = X_{t_i} + \frac{1}{2}\Delta t_i \cdot \hat{v}_{t_i}(X_{t_i})$

10:     $v_{t_i + \frac{1}{2}\Delta t_i}(X_{t_i + \frac{1}{2}\Delta t_i}) = v(X_{t_i + \frac{1}{2}\Delta t_i}, t_i + \frac{1}{2}\Delta t_i, \Phi(\text{prompt}); \varphi)$        {Run & Save to GPU Memory}

11:     $X_{t_{i-1}} = X_{t_i} + \Delta t_i \cdot v_{t_i + \frac{1}{2}\Delta t_i}(X_{t_i + \frac{1}{2}\Delta t_i})$

12: **end for**

13: **return** $X_0$

---

# B. Technical Proofs

This section provides detailed technical proofs for the theoretical results discussed in this paper.

## B.1. Proof of Proposition 3.1

*Proof.* The reverse ODE is given by:

$$\frac{\mathrm{d}x}{\mathrm{d}t} = -v(x, t). \tag{13}$$

Let $x^{\text{True}}(t)$ be the true solution of the reverse ODE, starting from $x_T^{\text{True}}$, and $x^{\text{Perturbed}}(t)$ be the solution starting from $x_T^{\text{Numerical}} = x_T^{\text{True}} + \Delta_T$. The initial condition difference is:

$$x_T^{\text{Perturbed}}(T) - x_T^{\text{True}}(T) = \Delta_T. \tag{14}$$

Define the error $\Delta(t)$ as the difference between the perturbed and true solutions:

$$\Delta(t) = x^{\text{Perturbed}}(t) - x^{\text{True}}(t). \tag{15}$$

The dynamics of $\Delta(t)$ follow from the reverse ODE:

$$\frac{\mathrm{d}\Delta(t)}{\mathrm{d}t} = \frac{\mathrm{d}x^{\text{Perturbed}}(t)}{\mathrm{d}t} - \frac{\mathrm{d}x^{\text{True}}(t)}{\mathrm{d}t}. \tag{16}$$

Substituting the reverse ODE for each term:

$$\frac{\mathrm{d}\Delta(t)}{\mathrm{d}t} = -v(x^{\text{Perturbed}}(t), t) + v(x^{\text{True}}(t), t). \tag{17}$$

Using the Lipschitz continuity of $v(x, t)$, we have:

$$\|v(x^{\text{Perturbed}}(t), t) - v(x^{\text{True}}(t), t)\| \le L\|\Delta(t)\|, \tag{18}$$

where $L$ is the Lipschitz constant of $v(x, t)$. Thus,

$$\left\|\frac{\mathrm{d}\Delta(t)}{\mathrm{d}t}\right\| \le L\|\Delta(t)\|. \tag{19}$$

Based on the definition of the derivative of a norm:

$$\frac{\mathrm{d}\|\Delta(t)\|}{\mathrm{d}t} = \frac{\Delta(t)}{\|\Delta(t)\|} \cdot \frac{\mathrm{d}\Delta(t)}{\mathrm{d}t} \le \left\|\frac{\Delta(t)}{\|\Delta(t)\|}\right\| \cdot \left\|\frac{\mathrm{d}\Delta(t)}{\mathrm{d}t}\right\| = \left\|\frac{\mathrm{d}\Delta(t)}{\mathrm{d}t}\right\| \le L\|\Delta(t)\|, \tag{20}$$

This can be rewritten as:

$$\frac{\mathrm{d}\|\Delta(t)\|}{\|\Delta(t)\|} \le L\,\mathrm{d}t. \tag{21}$$

Integrate both sides from $t = T$ (initial condition) to $t = 0$ (final condition):

$$\int_T^0 \frac{\mathrm{d}\|\Delta(t)\|}{\|\Delta(t)\|} \le \int_T^0 L\,\mathrm{d}t. \tag{22}$$

Thus, the inequality becomes:

$$\ln\|\Delta(0)\| - \ln\|\Delta(T)\| \le -LT. \tag{23}$$

Simplify:

$$\ln\|\Delta(0)\| \le \ln\|\Delta(T)\| - LT. \tag{24}$$

Exponentiate both sides to remove the logarithm:

$$\|\Delta(0)\| \le \|\Delta(T)\|e^{-LT}. \tag{25}$$

$$\square$$

## B.2. Proof of Proposition 4.1

*Proof.* Define the time interval between $t$ and $t - 1$ as $\Delta t$ for simplicity, the reused velocity at $t$ is given by:

$$\hat{v}_\theta(X_t, t) = v_\theta(X_{(t-1)+\frac{\Delta t}{2}}, (t - 1) + \frac{\Delta t}{2}), \tag{26}$$

where $X_{(t-1)+\frac{\Delta t}{2}}$ is computed recursively using:

$$X_{(t-1)+\frac{\Delta t}{2}} = X_{t-1} + \frac{\Delta t}{2}\hat{v}_\theta(X_{t-1}, t - 1). \tag{27}$$

The exact velocity at $t$ is:

$$v_\theta(X_t, t). \tag{28}$$

To quantify the difference $||\hat{v}_\theta(X_t, t) - v_\theta(X_t, t)||$, we expand the reused velocity $\hat{v}_\theta(X_t, t)$ around the exact velocity $v_\theta(X_t, t)$. Using a Taylor series expansion, expand $v_\theta(X_{(t-1)+\frac{\Delta t}{2}}, (t-1) + \frac{\Delta t}{2})$ around $X_t$ and $t$ :

$$v_\theta(X_{(t-1)+\frac{\Delta t}{2}}, (t-1) + \frac{\Delta t}{2}) \approx v_\theta(X_t, t) + \frac{\partial v_\theta}{\partial X}(X_{(t-1)+\frac{\Delta t}{2}} - X_t) + \frac{\partial v_\theta}{\partial t}(-\Delta t + \frac{\Delta t}{2}) + \mathcal{O}(\Delta t^2). \quad (29)$$

The temporal difference is:

$$-\Delta t + \frac{\Delta t}{2} = \frac{-\Delta t}{2}. \quad (30)$$

Thus, the temporal contribution to the velocity difference is:

$$-\frac{\Delta t}{2}\frac{\partial v_\theta}{\partial t}. \quad (31)$$

The leading term introduces an error of $\mathcal{O}(\Delta t)$.

The spatial difference is:

$$X_{(t-1)+\frac{\Delta t}{2}} - X_t. \quad (32)$$

Using the recursive relationship:

$$X_{(t-1)+\frac{\Delta t}{2}} = X_{t-1} + \frac{\Delta t}{2}\hat{v}_\theta(X_{t-1}, t-1), \quad (33)$$

and the local truncation error of Euler method

$$X_t \approx X_{t-1} + \Delta t \cdot v_\theta(X_{t-1}, t-1) + \mathcal{O}(\Delta t^2), \quad (34)$$

we subtract:

$$X_{(t-1)+\frac{\Delta t}{2}} - X_t = -\Delta t \cdot v_\theta(X_{t-1}, t-1) + \frac{\Delta t}{2}\hat{v}_\theta(X_{t-1}, t-1) + \mathcal{O}(\Delta t^2). \quad (35)$$

Simplify:

$$X_{(t-1)+\frac{\Delta t}{2}} - X_t = \frac{\Delta t}{2}(\hat{v}_\theta(X_{t-1}, t-1) - 2 \cdot v_\theta(X_{t-1}, t-1)) + \mathcal{O}(\Delta t^2). \quad (36)$$

Substitute this into the spatial term:

$$\frac{\partial v_\theta}{\partial X}(X_{(t-1)+\frac{\Delta t}{2}} - X_t) = \frac{\Delta t}{2}\frac{\partial v_\theta}{\partial X}(\hat{v}_\theta(X_{t-1}, t-1) - 2 \cdot v_\theta(X_{t-1}, t-1)) + \mathcal{O}(\Delta t^2). \quad (37)$$

Combine both temporal and spatial difference:

$$\hat{v}_\theta(X_t, t) - v_\theta(X_t, t) = \frac{\Delta t}{2}\frac{\partial v_\theta}{\partial X}(\hat{v}_\theta(X_{t-1}, t-1) - v_\theta(X_{t-1}, t-1)) - \frac{\Delta t}{2}(\frac{\partial v_\theta}{\partial t} + \frac{\partial v_\theta}{\partial X}v_\theta(X_{t-1}, t-1)) + \mathcal{O}(\Delta t^2). \quad (38)$$

Let $\delta_t = ||\hat{v}_\theta(X_t, t) - v_\theta(X_t, t)||$. From the above analysis and triangle inequality:

$$\delta_t \le \frac{\Delta t}{2}\delta_{t-1} + \mathcal{O}(\Delta t). \quad (39)$$

Unfold the recursion:

$$\delta_t \le \mathcal{O}(\Delta t) + \frac{\Delta t}{2}\mathcal{O}(\Delta t) + \left(\frac{\Delta t}{2}\right)^2 \mathcal{O}(\Delta t) + \cdots. \quad (40)$$

This is a geometric series with common ratio $\frac{\Delta t}{2}$, summing to:

$$\delta_t \le \mathcal{O}(\Delta t) \cdot \sum_{k=0}^{\infty}\left(\frac{\Delta t}{2}\right)^k = \mathcal{O}(\Delta t) \cdot \frac{1}{1 - \frac{\Delta t}{2}}. \quad (41)$$

For small $\Delta t$, $\frac{1}{1-\frac{\Delta t}{2}} \approx 1 + \frac{\Delta t}{2}$, so:

$$\delta_t = ||\hat{v}_\theta(X_t, t) - v_\theta(X_t, t)|| \le \mathcal{O}(\Delta t). \quad (42)$$

$\square$

### B.3. Proof of Theorem 4.2

*Proof.* Define the time interval as $\Delta t$ for simplicity, our modified midpoint method updates $X_{t+1}$ as:

$$X_{t+1} = X_t + \Delta t \cdot v_\theta\left(X_{t+\frac{\Delta t}{2}}, t + \frac{\Delta t}{2}\right), \tag{43}$$

where:

$$X_{t+\frac{\Delta t}{2}} = X_t + \frac{\Delta t}{2}\hat{v}_\theta(X_t, t). \tag{44}$$

Substituting $X_{t+\frac{\Delta t}{2}}$, the velocity term becomes:

$$v_\theta\left(X_{t+\frac{\Delta t}{2}}, t + \frac{\Delta t}{2}\right) \approx v_\theta(X_t, t) + \frac{\Delta t}{2}\frac{\partial v_\theta}{\partial t} + \frac{\Delta t}{2}\frac{\partial v_\theta}{\partial X}\hat{v}_\theta(X_t, t) + \mathcal{O}(\Delta t^2). \tag{45}$$

Under the assumption $||\hat{v}_\theta(X_t, t) - v_\theta(X_t, t)|| \leq \mathcal{O}(\Delta t)$, we write:

$$\hat{v}_\theta(X_t, t) = v_\theta(X_t, t) + \delta v, \tag{46}$$

where $||\delta v|| \leq \mathcal{O}(\Delta t)$. Substituting this into the velocity term expansion:

$$\frac{\partial v_\theta}{\partial X}\hat{v}_\theta(X_t, t) = \frac{\partial v_\theta}{\partial X}v_\theta(X_t, t) + \frac{\partial v_\theta}{\partial X}\delta v. \tag{47}$$

Since $||\delta v|| \leq \mathcal{O}(\Delta t)$, the additional term $\frac{\Delta t}{2} \cdot \frac{\partial v_\theta}{\partial X}\delta v$ contributes an error of $\mathcal{O}(\Delta t^2)$, which is of the same order as higher-order terms in the expansion. Thus, the velocity term becomes:

$$v_\theta\left(X_{t+\frac{\Delta t}{2}}, t + \frac{\Delta t}{2}\right) \approx v_\theta(X_t, t) + \frac{\Delta t}{2}\frac{\partial v_\theta}{\partial t} + \frac{\Delta t}{2}\frac{\partial v_\theta}{\partial X}v_\theta(X_t, t) + \mathcal{O}(\Delta t^2). \tag{48}$$

Substituting the expanded velocity back into the updated equation:

$$X_{t+1} = X_t + \Delta t \cdot \left(v_\theta(X_t, t) + \frac{\Delta t}{2}\frac{\partial v_\theta}{\partial t} + \frac{\Delta t}{2}\frac{\partial v_\theta}{\partial X}v_\theta(X_t, t) + \mathcal{O}(\Delta t^2)\right). \tag{49}$$

Simplify:

$$X_{t+1} = X_t + \Delta t \cdot v_\theta(X_t, t) + \frac{\Delta t^2}{2}\frac{\partial v_\theta}{\partial t} + \frac{\Delta t^2}{2}\frac{\partial v_\theta}{\partial X}v_\theta(X_t, t) + \mathcal{O}(\Delta t^3). \tag{50}$$

The exact solution to the ODE is:

$$X(t + \Delta t) = X(t) + \Delta t \cdot v_\theta(X_t, t) + \frac{\Delta t^2}{2}\frac{\partial v_\theta}{\partial t} + \frac{\Delta t^2}{2}\frac{\partial v_\theta}{\partial X}v_\theta(X_t, t) + \mathcal{O}(\Delta t^3). \tag{51}$$

The modified midpoint method's update matches the exact solution up to $\mathcal{O}(\Delta t^2)$, confirming that the local truncation error remains $\mathcal{O}(\Delta t^3)$. Since the local truncation error is $\mathcal{O}(\Delta t^3)$, the global error accumulates over $\mathcal{O}(1/\Delta t)$ steps, resulting in $\mathcal{O}(\Delta t^2)$. $\square$

## C. Empirical Convergence Rate

We now present a comparative analysis of reconstruction errors for different solvers, considering up to 60 Number of Function Evaluations (NFE). Our method exhibits rapid convergence to a consistently low reconstruction error, demonstrating its efficiency in minimizing reconstruction loss. The vanilla ReFlow model, implemented with a first-order Euler method, suffers from a slow convergence rate, implying a higher computational cost to achieve comparable accuracy. Notably, the RF-solver, which incorporates a second-order truncation error, displays a non-monotonic convergence behavior. Although it initially converges more quickly than vanilla ReFlow, the reconstruction error begins to increase after approximately 25 steps. This suggests that the second-order approximation in the RF-solver may introduce instability or over-fitting beyond this point, emphasizing the robustness and efficiency of our proposed method.

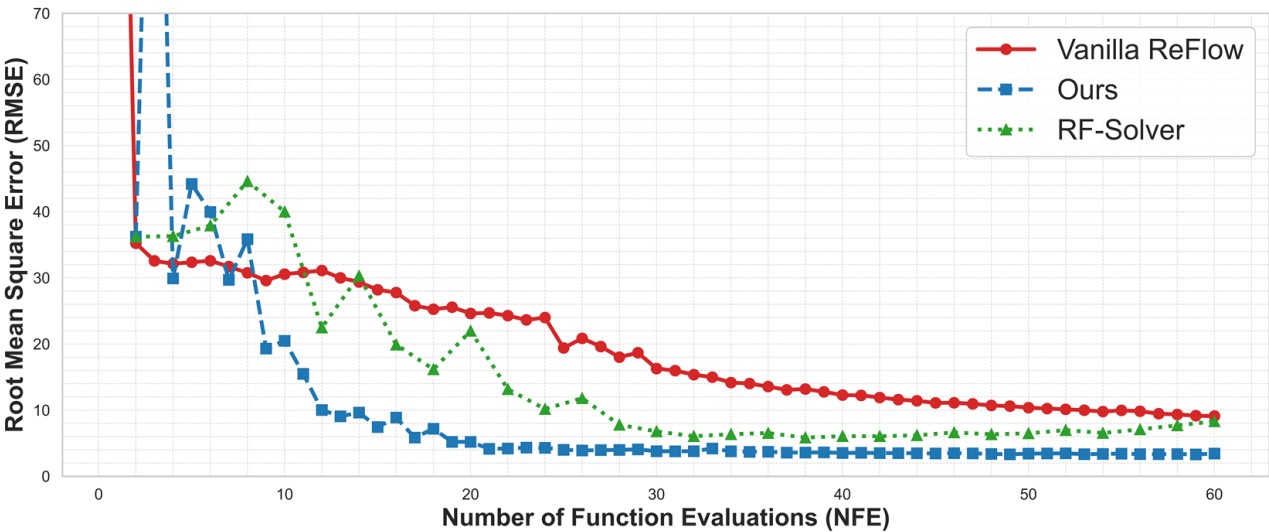

*Figure 7.* Visualization of the convergence rate of different order inversion and reconstruction method. With 60 NFE, our approach still enjoys the lowest reconstruction error and the fastest convergence speed.

## D. Python-style Pseudo-Code

In this section, we provide Python-style pseudo-code to elucidate the core algorithmic steps of our proposed approach. This approach, while conceptually simple, delivers substantial performance improvements. Specifically, initialization of the variable $\hat{v}$ is performed as None in the first iteration. Subsequently, each iteration requires only a single invocation of the model evaluation function, thereby maintaining a computational cost per iteration equivalent to that of the baseline ReFlow model employing the first-order Euler method. However, as evidenced by the convergence analysis presented in Figure 7, our approach achieves a significantly accelerated rate of convergence.

```python
hat_velocity = None
for t_curr, t_prev in zip(timesteps[:-1], timesteps[1:]):
    if hat_velocity is None:
        velocity = model(X, t_curr)
    else:
        velocity = hat_velocity
    X_mid = X + (t_prev - t_curr) / 2 * velocity
    velocity_mid = model(X_mid, t_curr + (t_prev - t_curr) / 2)
    hat_velocity = velocity_mid
    X = X + (t_prev - t_curr) * velocity_mid
return X
```

## E. Ablation Study

**Editing Steps:** An ablation study was conducted to evaluate the influence of the number of editing steps on the editing performance. The number of steps was varied from 2 to 12, and the corresponding results are presented in Figure 8. The analysis reveals that with only two editing steps, the editing prompts are not effectively incorporated, resulting in suboptimal performance. However, a significant improvement in performance is observed as the number of steps increases. Notably, the performance achieved with eight steps is comparable to that obtained with 10 or 12 steps, indicating that eight steps provide sufficient capacity for effective editing. Consequently, a value of eight editing steps was selected for use in all subsequent experiments.

**Inversion-free Editing:** Here, we apply our modified midpoint method to solve the controlled ODEs employed by other ReFlow-based editing methods. This compatibility underscores FireFlow's robustness in handling the ODE challenges inherent in ReFlow models, regardless of whether they occur during the forward or reverse diffusion process. Visual comparisons highlighting the superiority of our approach are shown in Figures 9.

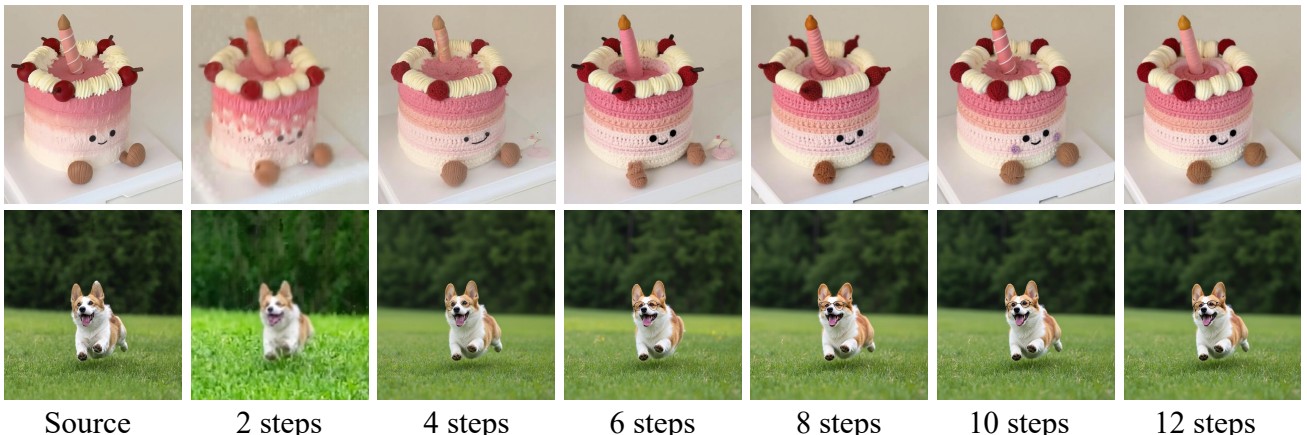

| Source | 2 steps | 4 steps | 6 steps | 8 steps | 10 steps | 12 steps |

*Figure 8.* Ablation Study on the Number of Editing Steps.

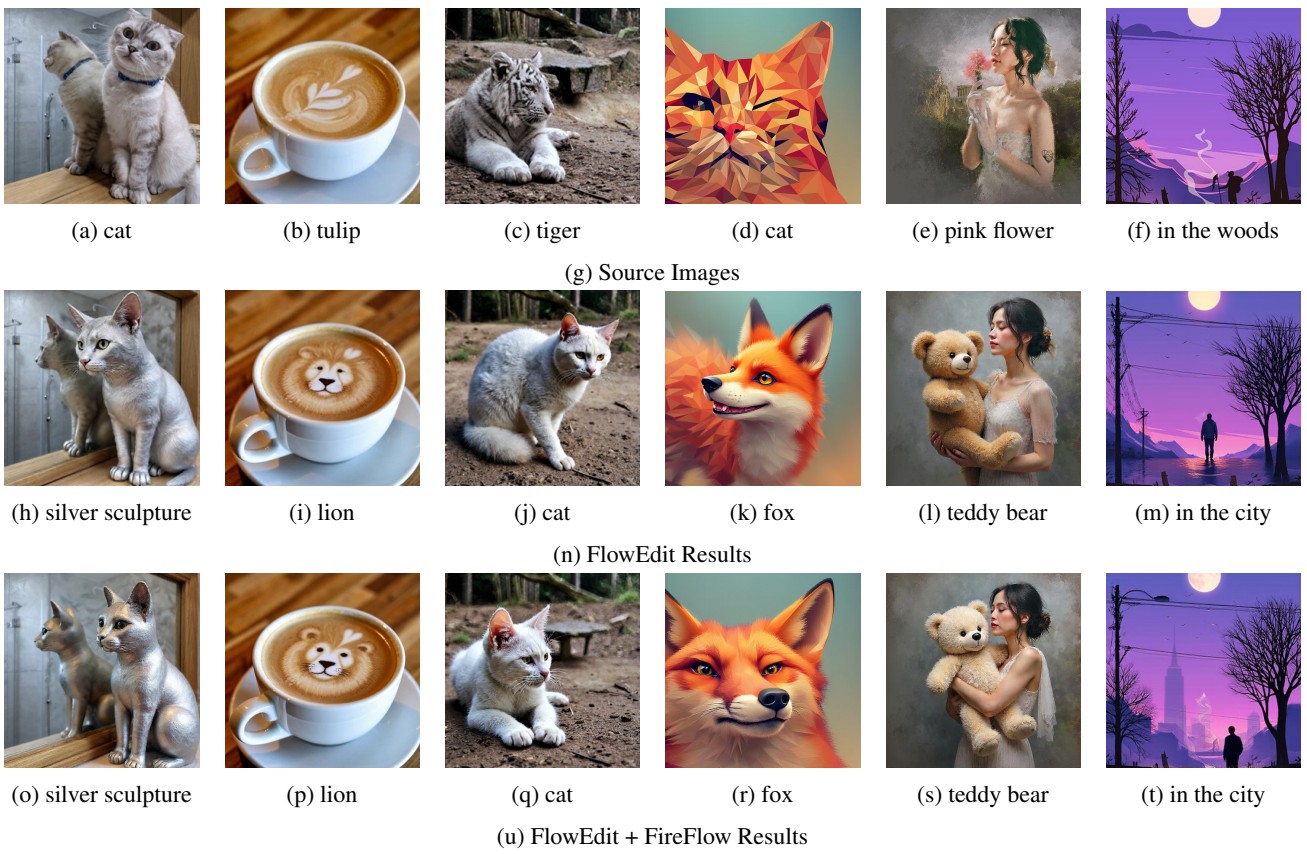

| (a) cat | (b) tulip | (c) tiger | (d) cat | (e) pink flower | (f) in the woods |

(g) Source Images

| (h) silver sculpture | (i) lion | (j) cat | (k) fox | (l) teddy bear | (m) in the city |

(n) FlowEdit Results

| (o) silver sculpture | (p) lion | (q) cat | (r) fox | (s) teddy bear | (t) in the city |

(u) FlowEdit + FireFlow Results

*Figure 9.* Illustration on the effect of FireFlow solver when coupled with other controlled ODE.

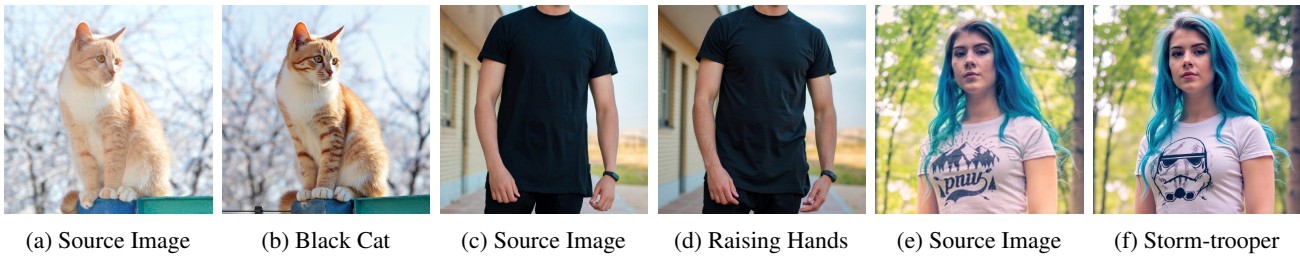

| (a) Source Image | (b) Black Cat | (c) Source Image | (d) Raising Hands | (e) Source Image | (f) Storm-trooper |

*Figure 10.* Illustrations on FireFlow Failure Cases.

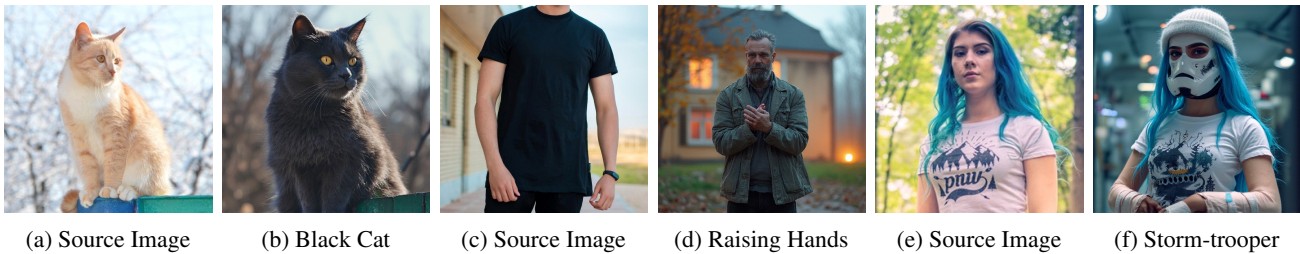

| (a) Source Image | (b) Black Cat | (c) Source Image | (d) Raising Hands | (e) Source Image | (f) Storm-trooper |

*Figure 11.* Illustrations on FireFlow with K feature addition in Self-attention.

## F. Limitations

We empirically observe that our approach struggles with editing tasks involving changes to object colors or uncommon scenarios in natural images. As illustrated in Figure 10, the cat's color remains unsatisfactory after editing. Similarly, in less common scenes, such as when the person's head is not visible in the image, the editing results are poor.

Another example involves an uncommon description, such as "a [stormtrooper] with blue hair wearing a shirt," which also yields unexpected results. We attribute these issues to the simplicity of the editing strategy, which involves only replacing the $V$ feature in the self-attention module:

$$\text{Self\_Attn}_{edit} = \text{Softmax}(\frac{Q_{edit}K_{edit}}{\sqrt{d}})V_{inv.} \tag{52}$$

This approach appears insufficient for handling such scenarios.

We empirically find that incorporating $K$ feature addition in the self-attention module can resolve these problems. Formally,

$$\text{Self\_Attn}_{edit} = \text{Softmax}(\frac{Q_{edit}(K_{edit} + K_{inv.})}{\sqrt{d}})V_{edit} \tag{53}$$

However, this comes at the cost of diminished preservation of the original structure and background details. Details can be found in Figure 11. We also include an quantitative analysis on different editing strategies in Table 7. The results are consistent with the illustrations. It is evident that Equation 53 belongs to the category of "cross-attention," focusing on merging features from the self-attention module during inversion with the corresponding features generated during the denoising process. This concept has been extensively discussed in diffusion model (DM) editing methods. Table 7 presents only the foundational attempts in this direction. We hope this will inspire further research and advancements in future work.

*Table 7.* Comparison on different editing methods. Results on PIE Bench are reported. Guidance terms indicate the guidance ratio settings used in the FLUX model during the denoising process.

| Method | Guidance | Structure | Background Preservation | | CLIP Similarity↑ | | Steps | NFE↓ |
|---|---|---|---|---|---|---|---|---|
| | | Distance↓ | PSNR↑ | SSIM↑ | Whole | Edited | | |
| Add $Q$ | [1,2,2,...] | 0.0590 | 17.72 | 0.7340 | 27.01 | 23.84 | **8** | **18** |
| Add $Q$ + Add $K$ | [1,2,2,...] | 0.0537 | 18.35 | 0.7520 | 26.82 | 23.60 | **8** | **18** |
| Add $Q$ + Add $K$ + Add $V$ | [1,2,2,...] | 0.0416 | 19.63 | 0.7805 | 25.95 | 22.92 | **8** | **18** |
| Add $Q$ | [1,1,2,...] | 0.0530 | 18.78 | 0.7580 | 26.99 | 23.85 | 15 | 32 |
| Add $Q$ + Add $K$ | [1,1,2,...] | 0.0486 | 19.30 | 0.7721 | 26.68 | 23.59 | 15 | 32 |
| Replace $V$ | [2,2,...] | **0.0271** | **23.03** | **0.8249** | 26.02 | 22.81 | **8** | **18** |
| Add $Q$ | [2,2,...] | 0.0710 | 16.49 | 0.7077 | **27.33** | **24.09** | **8** | **18** |
| Add $K$ | [2,2,...] | 0.0738 | 16.41 | 0.7066 | 27.25 | 24.01 | **8** | **18** |

