# OpenReview forum: "FireFlow: Fast Inversion of Rectified Flow for Image Semantic Editing"
_ICML.cc/2025/Conference — ICML 2025 poster_

### Official Review · Reviewer_DAG3 · 2025-03-10

**Overall Recommendation:** 3

**Summary:**

This work proposes a modified midpoint method, which is a well-known second order numerical solver. One drawback of midpoint method is that it needs an additional model evaluation for each sampling step, which makes the overall solver slow. This work instead proposes to piggyback on the previous model prediction in the first step of midpoint method. This reduces the number of model evaluations per step to 1. Surprisingly, the authors show that this modified midpoint method retains the  local error and global error of the second order methods while being computationally efficient. The authors demonstrate advantages of this approach in image editing tasks as well as in image reconstruction and inversion.

**Claims And Evidence:**

Some important quantitive experiments are missing an ablation/baseline — the original midpoint method. The proposed method modifies the original midpoint method with an approximation, but the tradeoffs between the two choices are difficult to infer from the provided quantitative results. Note that I understand that the proposed method needs only one NFE per sampling step as opposed to two NFEs of the original midpoint method, but there should be additional results as described below for a complete overview of the tradeoffs between the two:
    1. Table 4 should include quantitative results with the original midpoint method as well.
    2. For better understanding of the effects of modification of the first substep of the midpoint method, Figure 3 should ideally include error analysis of the original midpoint method. Similarly, Figure 4 and Figure 7 can also include RMSE values of the original midpoint method.

**Essential References Not Discussed:**

N/A

**Experimental Designs Or Analyses:**

The experiments considered in this work are sound.

**Methods And Evaluation Criteria:**

Yes, the proposed methods and evaluation methods make sense.

The paper considers PIE-Bench for editing tasks, and DCI (Densely captioned image) dataset for inversion and reconstruction tasks. The paper reports quantitative performance on standard metrics for T2I models such as PSNR, SSIM, FID, CLIP similarity etc.

**Other Comments Or Suggestions:**

1. In Figure 6, can the authors include the prompts that were provided for editing?
2. How many sampling steps were used to compute time cost in Table 5 for other methods besides Vanilla Reflow? Is this 8 steps for the proposed method and upto 28 steps for RF Solver? The exact number of steps will help understand the speedup better. Perhaps this can be indicated in an additional column or in the text. In addition, can we also include results for the original midpoint method to better demonstrate the inference time advantages (I presume it must be 2x of the proposed method)?
3. Minor (Clarification): Table 4 mentions NFEs for RF-Solver as 15 but the text (line 328) mentions it as 25.

**Other Strengths And Weaknesses:**

Strengths:
1. The proposed method is attractive as it provides the advantages of a second order method while being computationally more efficient.
2. The results included in the paper seem promising as this method seems to have both improved qualitative and quantitative performance compared to prior SOTA methods.

Weaknesses:
As mentioned above, the authors should also provide the ablation study against the original midpoint method for various applications to better understand the tradeoff between the two.

**Questions For Authors:**

1. In Figure 2, How were the pairs of points for plotting the transport maps/trajectories chosen? The number of points in the transport maps are fewer than the number of points in the scatter plot that shows the generated points from the target distribution. Also, could the authors elaborate why the trajectories of midpoint method are less straight then the proposed method (as the proposed method is expected to have a larger error constant, even though the error is of the same order)
2. Algorithms 1 and 2 mention $V_{t_{N-1}}^{inv}$ in self-attention layers. Could you elaborate the context of self-attention layer? Isn’t $V_{t_{N-1}}$ simply the prediction of the ReFlow network? How does this relate to the discussion in Lines 1008 - 1012 in the appendix?
3. Have the authors tried using the proposed method for unconditional image generation from ReFlow models? This will be an interesting and useful addition to the appendix.
4. Can the proposed method be used for solving SDEs similar to the original midpoint method? Do any of the assumptions in the proof not hold for SDEs (e.g. smoothness assumptions)? These limitations if any, should be discussed.

**Relation To Broader Scientific Literature:**

The proposed method seems to be general enough to also have implications beyond the specific task of semantic editing considered in this work which is in designing efficient numerical solvers for flow ODEs for unconditional image generation. However, this paper does not include this additional analysis as it is beyond the scope of the research problem considered in this paper, but it is an interesting direction nonetheless.

**Theoretical Claims:**

Yes. I checked the proof of Proposition 3.1, 4.1 and 4.2. The proofs seem correct.

---

> ### Author Rebuttal · Authors · 2025-03-31
>
> We sincerely appreciate the authors for carefully verifying our theoretical claims and constructive comments. Responses to your concerns as follow:
>
> Q1: Include the original midpoint method's results in Table 4 and its inference time in Table 5.
> - Although both ours and Midpoint are 2nd-order solvers, they exhibit different numerical stability when applied to ReFlow ODE. Since $v_\theta(x,t)$ predicted by neural network is not perfectly accurate, numerical solvers handle the resulting noise differently. Ours demonstrates robustness to these inaccuracies, leading to slightly better results.
>
> Solver|steps|time(s)|Structure Distance↓|PSNR↑|SSIM↑|CLIP Whole↑|CLIP Edited↑
> -|-|-|-|-|-|-|-
> Midpoint|15|24.6|0.0307|22.94|0.8208|25.93|22.88
> Midpoint|8|13.3|0.0318|22.45|0.8122|26.02|22.89
> Ours|8|7.7|0.0271|23.03|0.8249|26.02|22.81
>
> Q2: Fig. 3 should include error analysis of the original midpoint. Fig. 4 and 7 can include RMSE of the original midpoint.
> - Thanks for valuable suggestions. We hope to clarify that Fig. 3 illustrates the approximation error between the estimated midpoint velocity and true midpoint velocity. The original midpoint method does not involve such an approximation, so an error analysis is not applicable. Regarding Fig.7 with RMSE values, we revised the illustration of RMSE versus ODE steps "for better understanding of the effects of modification of the first substep of the midpoint method" at [link](https://ibb.co/mrBDMkTT).
>
> Q3: Can authors include the prompts used in Fig. 6?
> - Due to this year’s strict character limit for rebuttal, we hope to include the prompts in appendix to make room for addressing your other insightful comments. We appreciate your understanding.
>
> Q4: How many sampling steps were used to compute time cost in Table 5?
> - We used 8 steps for FireFlow, 15 steps for RF-Solver, and 28 steps for RF-Inversion, consistent with Table 4. We appreciate your suggestion and will include an additional column to indicate the number of steps.
>
> Q5: Table 4 mentions steps for RF-Solver as 15 but line 328 mentions it as 25.
> - We hope to clarify that line 328 states, “*up to 25* steps.” According to RF-Solver's official code, step number is not fixed and up to 25. Since 15 steps results in a computational cost comparable to RF-Inversion, we report 15 in Table 4 for a fair comparison.
>
> Q6: In Fig. 2, how were the pairs of points for plotting the transport maps/trajectories chosen? The number of points in transport maps are fewer than scatter plots.
> - We strictly follow the [official instructions](https://colab.research.google.com/drive/1CyUP5xbA3pjH55HDWOA8vRgk2EEyEl_P?usp=sharing) provided by the original ReFlow paper for visualizing Fig. 2. According to their code, the first 30 pairs of points are selected to draw the transport trajectory, while all pairs are used for the scatter plot.
>
> Q7: Could authors elaborate why trajectories of midpoint method are less straight than the proposed method?
> - Please kindly refer to Reviewer ekVQ’s Q3, where we explain why FireFlow performs better.
>
> Q8: Alg. 1 and 2 mention $V_{t_{N-1}}^{inv}$ in self-attention layers. Could you elaborate the context of self-attention layer? Isn’t $V_{t_{N-1}}$ simply the prediction of ReFlow model? How does this relate to the discussion in Lines 1008-1012?
> - We hope to clarify that $V_{t_{N-1}}^{inv}$ refers to the $V$ feature in $\text{softamx}(\frac{QK}{\sqrt{d}})V$ within the self-attention layer of ReFlow model, whereas $v_{t_{N-1}}(X_{t_{N-1}})$ denotes the prediction of network. Algorithms 1 and 2 mentioned $V_{t_{N-1}}^{inv}$ as it involved in the editing technique introduced by RF-Solver, which we follow to ensure a fair comparison  (detailed in “Image Semantic Editing” section). We observe that using $V_{t_{N-1}}^{inv}$ can lead to certain failure cases, and Lines 1008-1012 discuss its limitations along with some simple alternatives.
>
> Q9: Have authors tried using proposed method for unconditional image generation from ReFlow models?
> - Thanks for your suggestions. We follow the original ReFlow paper’s protocol for unconditional image generation on CIFAR10, using open-source 1-Rectified-Flow-distill weights. Our method performs well:
>
> Solver|steps↓|NFEs↓|FID↓|IS↑
> -|-|-|-|-
> Euler|15|15|5.67|9.20
> Euler|10|10|5.83|9.03
> Midpoint|5|10|5.45|9.27
> Ours|5|6|5.35|9.26
>
> Q10: Can FireFlow be used for solving SDEs similar to original Midpoint method? Do any assumptions in the proof not hold?
> - In fact, neither FireFlow nor the original Midpoint can be directly applied to SDEs because they do not properly handle the stochastic term. The noise term $dW_t$ in SDE representing a Wiener process is nowhere differentiable, rendering approximations that rely on smoothness invalid. Besides, if someone tries to use a midpoint method, would implicitly assume a Stratonovich interpretation, while SDEs are defined in the Itô sense. Standard ODE solvers don’t account for the correction term required when switching between these interpretations.

---

### Official Review · Reviewer_ekVQ · 2025-03-12

**Overall Recommendation:** 3

**Summary:**

This paper proposes a low-cost alternative method for a second-order ODE solver aimed at Rectified Flows. Compared to the common second-order ODE solver, which requires $2T$ NFEs, this method only needs $T+1$ NFEs while maintaining the sampling quality of the second-order ODE solver. The core idea is to replace $v_t$ in the standard midpoint method with the midpoint velocity $v_{(t-1)+\frac{\Delta t}{2}}$ from the previous $t-1$ step (Eq.8 vs. Eq.10), so that intermediate values from $t-1$ step can be cached during the denoising process and loaded in the t-th step, avoiding the need for 2 NFEs calculation at each step.

## Update After Rebuttal
The author's reply solved most of my problems, but I still feel that this work has some shortcomings in innovation, and I will keep my score.

**Claims And Evidence:**

This paper claims that the proposed method can maintain the sampling efficiency of a 1st-order ODE solver while achieving the sampling quality of a 2nd-order ODE solver. This claim is mainly verified through a comparison of its sampling quality-NFEs with those of the 1st-order Vanilla ReFlow and the 2nd-order RF-solver. The experimental comparison includes qualitative and quantitative results such as t2i, inversion then reconstruction, and inversion then prompt-guided editing.

According to the experimental results of the report and my understanding of the field, this claim has been verified.

**Essential References Not Discussed:**

Based on my understanding of the field, this paper discusses most methods of accelerating rectified flow by solving solvers.

**Experimental Designs Or Analyses:**

This paper first verifies the effectiveness of the method on synthetic mixed Gaussian data through a toy example, and then validates the effectiveness of the method for T2I, inversion-reconstruction and inversion-editing experiments.
From the perspective of accelerated generation without training through solving the ODEs solver, the experimental design is reasonable.

**Methods And Evaluation Criteria:**

This paper proposes a method that is meaningful for accelerating the sampling of rectified flows. The method is meaningful for accelerating rectified flow sampling, which helps to improve efficiency while ensuring the quality of sampling, thereby achieving fast inversion and prompt-guided editing.

**Other Comments Or Suggestions:**

Some chart captions are too simple, such as Table 2, Table 4, and Figure 6, which should have more complete and self-evident captions.

**Other Strengths And Weaknesses:**

Strengths: This paper relies on FLUX, a 12b t2i model, and it is meaningful to study its inversion acceleration.

Weaknesses: Some experimental validations are not sufficiently thorough. For example, as shown in Table 2, the effect of the 20-step method should be compared again to observe how much better its solver performance is compared to the synchronous number.

**Questions For Authors:**

1. Through the toy example in Figure 2, it can be found that the method proposed in this paper is better than the Midpoint method under the same NFE, but according to my understanding, the method proposed in this paper is a low-cost approximation of the Midpoint method. Why is the result even better than the Midpoint method?
2. In addition to the approximation of Eq.10~12 v_{t} and the introduction of cache mechanisms, what are the most valuable contributions of this paper, according to the author?
3. Under different seeds, what is the comparison of the results produced by inversion and editing, and do they have diversity?
4. When NFE is extremely small (1-4), how is the performance? Some distillation-based methods seem to be able to achieve inversion and generation with fewer steps.
5. Why does this article indeed compare with methods such as dpmsolver++, unipc, etc., and why can't these methods be adapted to the sampling method of rectified flow?
I will re-evaluate my score based on the author's reply.

**Relation To Broader Scientific Literature:**

The most relevant to this paper is the previous method RF-Solver [1]. RF-Solver has proven that expanding the first-order ODE solver of vanilla Rectified Flow into a second-order form will help improve the accuracy of inversion. The contribution of this paper lies in reducing the NFEs of the second-order ODE solver while maintaining the quality of sampling.


[1] Wang, J., Pu, J., Qi, Z., Guo, J., Ma, Y., Huang, N., Chen, Y., Li, X., and Shan, Y. Taming rectified flow for inversion and editing. arXiv preprint arXiv:2411.04746, 2024.

**Theoretical Claims:**

The core contribution of this paper lies in improving the algorithm in Eq.8~Eq.9 to Eq.10~Eq.12, where Eq.8~Eq.9 are the standard midpoint method.
By the way, the proof of proposition 3.1, 4.1, and Theorem 4.2 is also provided in the supplementary materials, providing a theoretical basis for the methods presented in this paper.

---

> ### Author Rebuttal · Authors · 2025-03-30
>
> We sincerely appreciate the reviewer’s detailed feedback and constructive suggestions for improving our paper. Below are our responses to your concerns:
>
> Q1: As shown in Table 2, the effect of the 20-step method should be compared again to observe how much better its solver performance is compared to the synchronous number.
> - Thank you for your valuable suggestion. We have now reported all three methods with the same 20 steps, and our approach continues to outperform the others, as shown below:
>
> Methods|Steps↓|NFE↓|FID↓|CLIP Score↑
> -|-|-|-|-
> FLUX-dev|20|20|26.77|31.44
> RF-Solver|20|40|25.54|31.39
> Ours|20|21|25.13|31.44
>
> Q2: Some chart captions are too simple, such as Table 2, Table 4, and Figure 6.
> - Thank you for pointing that out. We will revise the descriptions for Table 2, Table 4, and Figure 6 to make the captions more complete and self-explanatory.
>
> Q3: In Figure 2, why is the result of a low-cost approximation of the Midpoint method even better than the Midpoint method under the same NFE?
> - To clarify, the standard Midpoint method requires 2 NFEs per step, whereas our approach requires only 1 NFE per step. This means that, under the same computational cost, our method effectively has more steps. Using more steps (i.e., smaller step sizes) generally improves accuracy by reducing truncation error, which is bounded by the step size, in the discretization of the continuous ODE. This explains why our low-cost approximation achieves better results despite using the same total NFE.
>
> Q4: Except for the approximation, what are the most valuable contributions of this paper, according to the author?
> - We believe our work challenges the conventional notion that high-order ODE solvers must be computationally expensive. Beyond mere approximation, we introduce a new direction for accelerating ReFlow ODEs with a solid theoretical foundation and a fully training-free approach. Our method is distinctly different from training-based distillation and traditional model compression techniques. The modified Midpoint method simply serves as a starting point, we are actively exploring a broader group of high-order methods to further enhance efficiency.
>
> Q5: Under different seeds, what is the comparison of the results produced by inversion and editing, and do they have diversity?
> - We would like to clarify that, unlike T2I which begins from random noise, inversion-based editing and reconstruction start from a fixed original image as the initial point in the ReFlow ODE. As shown in Algorithm 1 and 2, solving the inversion and denoising ODE do not involve any randomness, resulting in no diversity across different seeds. For a quantitative analysis, please kindly refer to our response to Q1 for Reviewer MGiL.
>
> Q6: When NFE is extremely small (1-4), how is the performance? Some distillation-based methods seem to be able to achieve inversion and generation with fewer steps.
> - Thank you for your insightful question. Our current modified midpoint method is not designed for extremely small NFEs (1–4) and the results are unsatisfactory. However, we are actively exploring this avenue and have found higher-order solvers compatible with FireFlow, enabling faster inversion with as few as 4 steps, yet this exploration seems to extend beyond the scope of this submission.
> - Regarding distillation-based methods, FireFlow is a fully training-free approach, making direct comparisons with training-based methods less fair. Besides, efficient solver introduces a novel perspective for accelerating ReFlow models, offering a distinct and complementary direction to existing techniques.
>
> Q7: Why does this article indeed compare with methods such as dpmsolver++, unipc, etc., and why can't these methods be adapted to the sampling method of rectified flow?
> - DPM-Solver++ and UniPC rely on diffusion model ODEs, which contain an analytically tractable term. ReFlow lacks this structure, making these solvers ineffective. Specifically:
>   - Fast solvers for diffusion models exist because traditional diffusion trajectories are **curved** and inefficient, requiring precise numerical integration. In contrast, **ReFlow straightens these paths**. Since DPM-Solver++ and UniPC refine curved trajectories through multi-step corrections, they are unnecessary for ReFlow. Instead, as demonstrated by FireFlow, an efficient solver for ReFlow should focus on minimizing discretization error without requiring multiple steps.
>   - Formally, both DPM-Solver++ and UniPC leverage the diffusion model ODE structure $\frac{dx}{dt}=f(t)x+g(t)e\_\theta(x,t)$ where the linear term $f(t)x$ allows for **partial analytical integration**, enabling efficient numerical solvers. However, ReFlow follows a different ODE $\frac{dx}{dt} =v_\theta(x,t)$ which **lacks an explicit drift term** $f(t)x$. As a result, the analytical simplifications used by DPM-Solver++ and UniPC do not apply, making these fast solvers inapplicable without a fundamental redesign.

---

### Official Review · Reviewer_3efD · 2025-03-14

**Overall Recommendation:** 3

**Summary:**

The authors proposed a second-order ODE solver to speed up inversion and reconstruction of flow-based model. It reused saved midpoint velocity so the computation efficiency remains the same as first-order ODE. It is proven to be faster in speed and higher in quality, and also benefit image editing by a large margin.

**Claims And Evidence:**

Yes. The claims in the paper are proven with evidences.
- For example, the authors claimed that the inversion efficiency can be improved with the proposed second-order method. Compared with midpoint method, it reduces half of the NFEs.
- The authors claims the accuracy remains similar to standard midpoint methods. It is proven in the appendix given some constraints.
- The authors also showed results on PIE-bench and visual results to prove the editing quality.

**Essential References Not Discussed:**

Not found.

**Experimental Designs Or Analyses:**

The experiments have no obvious flaw. One concern is all the methods are evaluated using FLUX model which is somehow well trained. So it is not clear whether the method can be well generalized to other ReFlow models, especially when v is not accurate enough.

**Methods And Evaluation Criteria:**

Yes.

**Other Comments Or Suggestions:**

See above.

**Other Strengths And Weaknesses:**

See above.

**Questions For Authors:**

See above.

**Relation To Broader Scientific Literature:**

I am not confident that this method is novel enough. First, it is a fully training-free method which does not involve 'learning' methods which are better aligned with ICML;  Second, the 2nd-order or 3rd-order ODE can be also achieved by Numerical Extrapolation (using the previous two velocity values from the network to extrapolate next midpoint) which also only needs 1 NFE. Some proper comparison might be needed to mitigate the concerns of large delta_t and inaccurate v estimation. Third, the paper does not introduce enough novelty in inversion and editing. The proposed method remains having problems like all other inversion-based method like poor identity preservation.

**Theoretical Claims:**

The main theoretical claims are proposition 4.1 and theorem 4.2. They are intuitively correct, however, they may have some potential issues to the best of my knowledge.

- I am not sure whether the proposed method will downgrade to 1st-order ODE if the velocity estimation of the midpoint is not accurate enough, saying v is not well trained.
- Whether it will be better to estimate the next midpoint velocity by measuring the current velocity and previous velocity to propagate, instead of fully relaying on the velocity estimation?

But given the constraints and assumption, the proof is sound and correct.

---

> ### Author Rebuttal · Authors · 2025-03-30
>
> We sincerely appreciate the authors for carefully verifying our theoretical claims, especially given the heavy review process. We will do our best to address your concerns.
>
> Q1: It is a fully training-free method which does not involve 'learning' methods which are better aligned with ICML.
> - We respect the reviewer’s opinion on whether a fully training-free method fits within the scope of machine learning. However, we believe that generative models are a prominent topic in the field of machine learning, and the ReFlow model, as a pioneering work in this area, warrants attention. FireFlow, with its efficiency gains, demonstrates practical applicability, which aligns with the interests of the machine learning community, hence aligns with ICML.
>
> Q2: Whether it will be better to estimate the next midpoint velocity by measuring the current velocity and previous velocity to propagate, achieved by Numerical Extrapolation (using the previous two velocity values from the network to extrapolate next midpoint)?
> - Thank you for your insightful suggestion. If we understand correctly, the proposed numerical extrapolation involves using the previous two velocity values, $v_t$ and $v_{t-1}$, generated by the network to approximate the next midpoint velocity as $\hat{v}\_{t+\frac{1}{2}\Delta t}=v_t + 0.5(v_t-v_{t-1})$ and updating the state as $X_{t+1}=X_t+\Delta t\cdot\hat{v}_{t+\frac{1}{2}\Delta t}$. To evaluate this approach, we have conducted an ablation study on step selection, and our approach continues to outperform the others, with results on PIE-Bench presented below:
>
> Method|steps↓|NFEs↓|Structure Distance↓|PSNR↑|SSIM↑|CLIP Whole↑|CLIP Edited↑
> -|-|-|-|-|-|-|-
> Numerical Extrapolation|15|30|0.0398|22.48|0.7913|**26.14**|22.64
> Numerical Extrapolation|8|16|0.0658|21.45|0.7504|26.04|22.32
> Ours|**8**|**18**|**0.0271**|**23.03**|**0.8249**|26.02|**22.81**
>
> Q3: Whether the proposed method will downgrade to 1st-order ODE if the velocity estimation of the midpoint is not accurate enough, whether the method can be well generalized to other ReFlow models, especially when $v$ is not accurate enough.
> - Thank you for your insightful question. We recognize that the robustness of our approach, particularly when velocity estimation is less accurate, is an important aspect to explore further. Since the accuracy of $v$ directly impacts image quality, we assess our method on earlier-generation, medium-sized Stable Diffusion models, such as SD3-medium, which has a significantly lower ELO score than the FLUX model and serves as a meaningful test case. We apply different ODE solvers to ReFlow-based editing, and the results on PIE-Bench are as follows:
>
> Method|Order|steps↓|NFEs↓|Structure Distance↓|PSNR↑|SSIM↑|CLIP Whole↑|CLIP Edited↑
> -|-|-|-|-|-|-|-|-
> RF-Inversion|1st-order|28|56|0.0464|19.75|0.6951|25.20|**23.16**
> Ours|2nd-order|**8**|**18**|0.0476|19.41|0.6871|25.16|23.11
> Ours|2nd-order|15|32|**0.0439**|**20.07**|**0.7019**|**25.30**|23.07
>
> - While the speedup rate is lower than FLUX, our method consistently outperforms 1st-order ODE across most metrics, achieving approximately a 2× speedup while maintaining high-quality results. Given that the accuracy of $v$ is directly linked to image quality, we believe it is more meaningful to focus on state-of-the-art models with higher accuracy and better generation capabilities—such as the FLUX model, as discussed in our paper.
>
> Q4: The paper does not introduce enough novelty in inversion and editing, with problems like all other inversion-based method like poor identity preservation.
> - We respectfully disagree with the claim that our work lacks novelty in inversion and editing. Our fast ODE solver for ReFlow introduces a fundamentally new approach to accelerating generative models, with no direct overlap with existing acceleration methods. This is substantiated by both rigorous theoretical analysis and extensive experiments, demonstrating clear improvements.
> - As with any training-free editing method, certain failure cases are expected, and claiming otherwise would be unrealistic. This is precisely why we have transparently acknowledged limitations in the appendix. However, occasional challenges in identity preservation should not overshadow the consistently strong performance demonstrated on widely recognized benchmarks. Our results indicate that FireFlow surpasses prior methods in efficiency and effectiveness.
> - More importantly, our core contribution extends beyond individual editing performance—it establishes a new direction for accelerating ReFlow models. This is not merely an incremental improvement but a principled advancement, supported by both theoretical foundations and empirical validation.

---

> > ### Comment · Reviewer_3efD · 2025-04-07
> >
> > Thanks for the detailed rebuttal. For the numerical extrapolation, it is better to consider the timestep difference but not simply averaging them. I believe the conclusions will be slightly different and encourage the authors to add them in the paper.
> > I will upgrade the scores to weak accept because the authors tried the methods on other base models in the rebuttal and proves its effectiveness. But I still hope AC can balance the novelty and technical depth in the final evaluation. The authors did a great work on proposing a simple yet effective inversion method, evaluate its effectiveness and efficiency, and prove the quality on a wide range of editing benchmark. But it has little impact on learning approach, and is also not tightly relevant to improving core image editing quality. Also the truth that 2nd-order ODE helps on sampling is not novel, although the authors used a similar trick to numeric extrapolation to reduce NFEs.

---

> > > ### Author Response · Authors · 2025-04-08
> > >
> > > We sincerely thank the reviewer for the thoughtful reconsideration of our work and for providing constructive suggestions, especially regarding the numerical extrapolation and the discussion on novelty and technical depth. Please find our response below, which we hope will be viewed as a continuation of academic discussion rather than self-defense.
> > >
> > > - Following your helpful suggestion, we revisited the numerical extrapolation by incorporating timestep differences rather than simply averaging. If we understand correctly, the reviewer suggests estimating the midpoint velocity using a weighted formulation such as
> > > $\hat{v}\_{t+\frac{1}{2}\Delta t} = \alpha \cdot v_t + (1 - \alpha) \cdot v_{t-1}$,
> > > where $\alpha$ is a weight determined by the timestep difference. A straightforward implementation is
> > > $\alpha = \frac{\Delta t_{\text{current}}}{\Delta t_{\text{current}} + \Delta t_{\text{prev}}}$,
> > > which assigns more weight to the current timestep when it is larger, reflecting the intuition that more recent velocities are more informative. We have conducted experiments using this formulation, and the updated results will be included in the revised version of the paper.
> > >
> > > Method|steps|NFEs|Structure Distance↓|PSNR↑|SSIM↑|CLIP Whole↑|CLIP Edited↑
> > > -|-|-|-|-|-|-|-
> > > Numerical Extrapolation|15|30|0.0397|22.44|0.7911|26.12|22.70
> > > Ours|8|18|0.0271|23.03|0.8249|26.02|22.81
> > > - We also sincerely appreciate the reviewer’s comments on the balance between novelty and technical depth. We fully agree with and respect this perspective. At the same time, we would like to slightly clarify the motivation and contribution of our work. Forward and inverse ODEs are central to ReFlow-based editing methods, which recently represent a mainstream direction in image editing. Our contribution lies in providing a fast and effective method that contributes to solving ReFlow ODEs, making our work relevant within this framework. Compared to the numerical extrapolation baseline, our method provides clear empirical and theoretical improvements in reducing NFEs, and is non-trivial, introducing a principled mechanism rather than a simple variant of existing tricks. Beyond the basic application of second-order solvers, our goal is to challenge the conventional belief that higher-order methods must be computationally expensive. By designing a lightweight yet effective scheme tailored to the characteristics of ReFlow ODEs, we hope to inspire future research on efficient solver designs for image generation tasks.
> > >
> > > Once again, we thank the reviewer for the valuable feedback and for the encouraging remarks on our work.

---

### Official Review · Reviewer_MGiL · 2025-03-14

**Overall Recommendation:** 4

**Summary:**

This paper introduces ​FireFlow, a fast inversion and editing method for Rectified Flow (ReFlow) models, designed to enable accurate image reconstruction and semantic editing with minimal computational overhead, enabling fast, high-fidelity image editing while fully leveraging the model's inherent linear motion properties. The approach advances practical applications of ReFlow-based generative models without architectural modifications or training.

**Claims And Evidence:**

The claims are ​largely supported by theoretical analysis and extensive experiments. The core contributions (efficient high-order solver, empirical superiority) are valid and impactful. Some suggestions for revisions:
The evaluation on PIE-Bench uses a fixed seed; variance across multiple runs is not reported.
Involving humen assessment would strength the contribution.

**Essential References Not Discussed:**

N/A

**Experimental Designs Or Analyses:**

Most results are compelling, several aspects of the experimental design and analysis warrant closer scrutiny:
- The baselines (e.g., RF-Solver, RF-Inversion) are compared at different step counts and NFEs. Without controlling for NFEs or steps, the claimed speedup (e.g., "3× runtime speedup") may exaggerate the method's efficiency.

**Methods And Evaluation Criteria:**

The proposed methods and evaluation criteria are well-aligned with the problem of efficient inversion and semantic editing for ReFlow-based generative models.  Evaluation criteria are ​appropriate and comprehensive, covering quality, accuracy, and speed. Minor gaps in dataset diversity and human evaluation could be addressed in future work but do not undermine the core contributions. ​

**Other Comments Or Suggestions:**

Line 72: This motivates a closer investigation. The format of Table 3 could be revised.

**Other Strengths And Weaknesses:**

Strengths
- The authors propose a modified midpoint method that achieves ​second-order accuracy with first-order computational cost, a creative adaptation of ODE solvers tailored for ReFlow's constant velocity dynamics. This is a significant departure from prior ReFlow inversion methods (e.g., RF-Solver, RF-Inversion) that either sacrifice accuracy for speed or require higher NFEs.
- Establishes error bounds for velocity reuse (Proposition 4.1) and proves global truncation error equivalence to standard midpoint methods (Theorem 4.2), advancing the theoretical understanding of ReFlow inversion.
- Logical flow from motivation to theory, method, and experiments are well-structured. Algorithms 1–2 and synthetic 2D experiments clearly illustrate the solver's mechanics. Code availability and detailed ablation studies (e.g., step-size vs. error in Figure 3) enhance reproducibility.

Weaknesses
- The statistics are reported with fixed seed.
- More discussions on limitations should be given. Experiments focus on ​semantic edits (e.g., object replacement). Complex edits like style transfer, multi-object manipulation is unexplored, limiting the perceived versatility of the method.

FireFlow’s ​innovative solver design and ​strong empirical results make it a compelling contribution to fast inversion/editing in ReFlow models. While limited comparisons and narrow task scope slightly weaken the narrative, its theoretical rigor, efficiency gains, and practical applicability position it as a valuable advancement for generative ODEs.

**Questions For Authors:**

How to enhance the proposed method for more challenging editing tasks?

Can this method be extrapolated to further accelerate by approximating higher-order methods through more caching intermediate steps?

**Relation To Broader Scientific Literature:**

The key contributions of FireFlow build upon and extend several lines of research in generative modeling, numerical methods for ODEs, and image inversion/editing, would benefit researchers in related fields.

**Theoretical Claims:**

The numerical experimental results and theoretical results are not perfect matches, there are slight inconsistencies at some points, but I agree with the author that the overall trend is consistent. The findings presented in this paper contribute to this field.

---

> ### Author Rebuttal · Authors · 2025-03-30
>
> Q1: Results on PIE-Bench are reported with fixed seed? Variance across multiple runs is not reported.
> - Thank you for your valuable feedback. To clarify, T2I generation starts from random noise determined by seed, while inversion-based editing starts from a fixed image as the initial point in ODE and solving ODE does not involve randomness. Nonetheless, we have re-run the experiments 5 times using different random seeds:
>
> PIE-Bench|Structure Distance↓|PSNR↑|SSIM↑|CLIP Whole↑|CLIP Edited↑
> -|-|-|-|-|-
> Reported in the paper|0.0271|23.03|0.8249|26.02|22.81
> 5-runs Mean|0.0271|23.03|0.8249|26.02|22.81
> 5-runs Std|0.|0.|0.|0.|0.
>
> Q2: Minor gaps in dataset diversity and human evaluation.
> - Thank you for your comment. We evaluated FireFlow using three public datasets—MSCOCO Caption, Densely Captioned Images (DCI), and PIE-Bench—across T2I, inversion, and editing tasks. While additional experiments could further support our findings, the current results already show significant improvements over recent methods. Notably, PIE-Bench includes 10 diverse editing tasks, including style transfer, as requested.
> - Regarding human evaluation, we are conducting a crowdsourced blind experiment. Due to time constraints, we couldn’t gather enough voting results for the rebuttal. However, we will include the user study results in the appendix for a more comprehensive evaluation.
>
> Q3: Without controlling for NFEs or steps for baseline methods, will the claimed speedup (e.g., "3× runtime speedup") be exaggerated?
> - Thank you for your insightful question. To clarify, we did not intentionally increase the NFEs or steps in the baseline methods to exaggerate the reported speedup. We followed the standard settings for a fair comparison. Besides, we conducted an ablation study on RF-Inversion and RF-Solver, which shows that reducing the number of NFEs or steps leads to higher image editing failure rates. This demonstrates that the selected steps are more optimal for editing tasks.
>
> Method|steps|NFEs|Structure Distance↓|PSNR↑|SSIM↑|CLIP Whole↑|CLIP Edited↑|Qualitative Analysis
> -|-|-|-|-|-|-|-|-
> RF-Inv.|28|56|0.0406|20.82|0.7192|25.20|22.11|Better trade-off
> RF-Inv.|14|28|*0.0677*|*18.88*|*0.7160*|26.03|23.20|**Failed to preserve original content**|
> RF-Solver|15|60|0.0311|22.90|0.8190|26.00|22.88|Better trade-off|
> RF-Solver|8|32|0.0211|24.93|0.8597|*24.47*|*21.25*|**Failed to conduct editing**
>
> Q4: More discussions on the cause of limitation should be given. Any solutions for these cases?
> - Thank you for your valuable comment. As a fast, training-free inversion-based editing method, we believe the limitations of our approach can be attributed to the following factors, along with potential solutions:
>   - Practical conditions vs. Proposition 4.1: As shown in Table 4, increasing the number of steps contributes to smoother velocity, which in turn improves performance. Additionally, more powerful ReFlow models offer more accurate velocity estimations, helping to bridge the gap between the theoretical assumptions and real-world scenarios. This is reflected in Table 7, where SD3.5 performs better than SD3.
>   - Editing technique compatibility: Our method works as a fast solver for ReFlow models, which makes it compatible with other ReFlow-based editing techniques. However, it inherits both the strengths and weaknesses of those methods. As such, improving the design of the editing technique itself will further enhance the performance of our approach.
>   - Prompt suitability: It is well-known that prompt engineering plays a crucial role in image generation. The official prompt for PIE-Bench is relatively brief, which could limit further improvements in editing quality. A more sophisticated vLLM-based prompt generator could better align the prompt with the editing task and yield better results.
>
> Q5: Experiments focus on ​semantic edits (e.g., object replacement). Complex edits like stylization, multi-object manipulation is unexplored.
> - Thank you for pointing that out. We apologize for the bias in the illustrations, which mainly focus on object replacement and may have been misleading. In fact, our approach performs well on more complex tasks, including stylized generation and multi-object manipulation. For additional results, please refer to the following [anonymous link](https://ibb.co/F4XzBrbM).
>
> Q6: Can this method be extrapolated to further accelerate by approximating higher-order methods through more caching intermediate steps?
> - Thank you for your insightful suggestion. We are actively exploring this avenue and have already found higher-order solvers compatible with FireFlow, enabling faster inversion with as few as 4 steps. We are also working towards a theoretical proof that a class of numerical solvers can be accelerated in this way.
> - However, we believe the current submission is self-contained, and we view this as an exciting direction for future work. We appreciate your suggestion and plan to explore it further in subsequent research.

---

### Decision · Program_Chairs · 2025-05-01

**Decision:**

Accept (poster)

**Comment:**

All reviewers support the acceptance of the paper, with one Accept and three Weak Accepts. The main contribution is a numerical solver for the ODEs in RectifiedFlow models, which accelerates model inversion and reconstruction. The contribution is thus closely tied to this specific class of models. As the reviewers acknowledge the benefits of the proposed method, I recommend acceptance. The authors are encouraged to incorporate the points raised during the rebuttal into the final version.